

# Drivers of 21st Century carbon cycle variability in the North Atlantic Ocean

Matthew P. Couldrey[1,a], Kevin I. C. Oliver[1], Andrew Yool[2], Paul R. Halloran[3], and Eric P. Achterberg[1,4]

[1]Ocean and Earth Science, University of Southampton, National Oceanography Centre, Southampton, UK
[2]National Oceanography Centre, Southampton, UK
[3]Geography, College of Life and Environmental Sciences, University of Exeter, UK
[4]GEOMAR Helmholtz-Zentrum für Ozeanforschung, Kiel, Germany
[a]Now at: Department of Meteorology, University of Reading, Reading, UK

**Correspondence:** Matthew P. Couldrey (couldrey.matthew@gmail.com)

**Abstract.** The North Atlantic carbon sink is a prominent component of global climate, storing large amounts of atmospheric carbon dioxide ($CO_2$), but this basin's $CO_2$ uptake variability presents challenges for future climate prediction. A comprehensive mechanistic understanding of the processes that give rise to year-to-year (interannual) and decade-to-decade (decadal) variability in the North Atlantic's dissolved inorganic carbon (DIC) inventory is lacking. Here, we numerically simulate the oceanic response to human-induced (anthropogenic) climate change from the industrial era to the year 2100. The model distinguishes how different physical, chemical, and biological processes modify the basin's DIC inventory; the saturation, soft tissue, and carbonate pumps, anthropogenic emissions, and other processes causing air-sea disequilibria. There are four 'natural' pools (saturation, soft tissue, carbonate, and disequilibrium), and an 'anthropogenic' pool. Interannual variability of the North Atlantic DIC inventory arises primarily due to temperature- and alkalinity-induced changes in carbon solubility (saturation concentrations). A mixture of saturation and anthropogenic drivers cause decadal variability. Multidecadal variability results from the opposing effects of saturation versus soft tissue carbon, and anthropogenic carbon uptake. By the year 2100, the North Atlantic gains 66 Pg (1 Pg = $10^{15}$ grams) of anthropogenic carbon, and the natural carbon pools collectively decline by 4.8 Pg. The first order controls on interannual variability of the North Atlantic carbon sink size are therefore largely physical, and the biological pump emerges as an important driver of change on multidecadal timescales. Further work should identify specifically which physical processes underlie the interannual saturation-dominated DIC variability documented here.

*Copyright statement.* TEXT

# 1 Introduction

The carbon cycle is the system of exchanges of carbon between and within atmospheric, oceanic, terrestrial biospheric and lithospheric reservoirs. The distribution of the planet's carbon between these reservoirs affects (and is affected by) the Earth's climate, forming a complex network of exchanges and feedbacks. Human activities, namely fossil fuel combustion, cement



production, and land use changes over the latest three centuries have perturbed this system from its preindustrial state (Ciais et al., 2013). Energy and cement production have rapidly released carbon from the lithosphere into the other three reservoirs, while the changing use of land causes a net removal of carbon from the terrestrial biosphere. One of the main effects of the addition of so-called 'anthropogenic carbon' to the atmosphere is to alter the radiative balance of the planet, which then has

a range of interconnected knock-on effects on Earth's climate and carbon cycle. However, less than half of the anthropogenic carbon emitted to the atmosphere remains there; ∼25% of these emissions is absorbed by the oceans and ∼30% by the terrestrial biosphere (Ciais et al., 2013; Le Quéré et al., 2016). The importance of the oceans for the carbon cycle and the climate is well known, yet the details of its role are still to be understood. In particular, the variability of the size of the ocean carbon reservoir over interannual to multidecadal timescales is not well understood. This variability in oceanic carbon concentrations is a

leading-order control on the amount of $CO_2$ that the oceans absorb (Doney et al., 2009; McKinley et al., 2011; Couldrey et al., 2016).

## 1.1 Components of the ocean carbon reservoir

In the atmosphere, the main form of gaseous carbon is carbon dioxide ($CO_2$), and its concentration is often expressed as a partial pressure of the total gas mixture, $pCO_2$. A sample of seawater in contact with the atmosphere will, given time,

reach an equilibrium or saturation concentration of dissolved inorganic carbon (DIC) with respect to the atmospheric carbon concentration. The magnitude of this seawater saturation concentration is set by the temperature ($T$), salinity ($S$) and total alkalinity ($A_T$) of seawater (and other dissolved species also play minor roles) (Weiss, 1974; Wolf-Gladrow et al., 2007). The vast majority of DIC present in the global ocean results from the partial or complete equilibration of seawater with the carbon dioxide present in the preindustrial atmosphere (Williams and Follows, 2011). This means that to first order, a water parcel's

DIC concentration is primarily set while it is in contact with the atmosphere in the mixed layer, based on atmospheric $pCO_2$, seawater temperature, salinity and alkalinity.

The timescale required for the mixed layer to equilibrate with atmospheric $CO_2$ levels is on the order of 0.5-1.5 years (Broecker and Peng, 1974; Jones et al., 2014). This timescale is often longer than the flushing time of the mixed layer, and so water may leave the mixed layer before it has entirely equilibrated with atmospheric $CO_2$. Furthermore, air-sea $CO_2$ dis-

equilibria are also generated by physical, chemical and biological processes acting on faster timescales than the equilibration timescale of $CO_2$. Photosynthetic fixation of $CO_2$, respiration of organic matter, and mixed layer depth physical variability may all act to modify local DIC concentrations away from atmospheric equilibria, while the buffering effect of seawater carbonate chemistry slows equilibration relative to nonreactive gases (Jones et al., 2014). Seawater DIC concentrations are further modified away from saturation concentrations once a water parcel departs the mixed layer into the ocean interior by two main

processes: the soft-tissue pump and the carbonate (or hard-tissue) pump.

The soft tissue pump is a keystone biogeochemical process by which organic matter is moved from the upper ocean to the interior. In the top, sunlit hundred or so meters of ocean, phytoplanktonic primary producers take up dissolved inorganic carbon and nutrients. When these organisms die, some portion of their constituent organic matter on the order of 10–35% (Buesseler and Boyd, 2009; Cavan et al., 2017) sinks into the deep ocean as particulates. Here, heterotrophs consume this sinking organic



matter, and release $CO_2$, nitrogen-, silicon-, and phosphorus-bearing waste back into solution. This whole process 'pumps' approximately 11 Pg C $yr^{-1}$ (1 Pg = $10^{15}$ grams) out of the photic surface zone (Sanders et al., 2014) and causes an enrichment of DIC beneath the mixed layer, characteristically below 300 m in the open ocean (Gruber and Sarmiento, 2002). Water that has been recently ventilated (such as North Atlantic Deep Water in the Atlantic) tends to show lower concentrations of DIC than

older waters (such as the deep waters of the Pacific) because in the latter, the soft tissue pump has caused carbon to accumulate with water mass age (Williams and Follows, 2011).

In addition to the soft tissue pump, sinking particles can also enrich deep waters with carbon through the carbonate (or hard tissue) pump. In the photic surface ocean, calcifying plankton such as coccolithophorids produce calcium carbonate ($CaCO_3$) shells. When these shells sink into the deep ocean, many dissolve back into solution, adding carbon to the DIC pool, while

the remainder sinks to the ocean floor and becomes buried through sedimentation. Much like with the soft tissue pump, the carbonate pump adds carbon to water masses over time, causing DIC enrichment in older waters. However, the carbonate pump tends to increase ocean DIC at greater depths than the soft tissue pump because the depths at which seawater is sufficiently corrosive to $CaCO_3$ are deeper than soft tissue remineralization depths (Williams and Follows, 2011).

Recent decades of ocean carbon cycle research have yielded a useful understanding of the currently observed, large scale,

time-mean ocean carbon reservoir, while highlighting important areas still in need of development (Takahashi et al., 2014; Key et al., 2015; Bakker et al., 2016; Olsen et al., 2016). The total oceanic inorganic carbon inventory is on the order of 38,000 Pg C (Key et al., 2004, 2015; Olsen et al., 2016). On the order of 90% of this inventory is accountable in terms of saturation with the preindustrial atmosphere (Williams and Follows, 2011). The DIC added to seawater at depth by the soft tissue pump contributes approximately 7% of the total budget (Volk and Hoffert, 1985). A few percent of the total DIC budget derives from

the carbonate pump (Williams and Follows, 2011). The total amount of DIC that exists due to subduction of water masses with considerable disequilibrium carbon is not well defined. The final, anthropogenic component of ocean DIC of 155±30 Pg C (Ciais et al., 2013) represents approximately half a percent of the total. The current, instantaneous or mean state of the ocean carbon reservoir is therefore relatively better quantified (although still crucially incomplete) than its variability in time.

## 1.2 North Atlantic Carbon Sink Variability

Fixed point observatories provide a comprehensive assessment of biogeochemical changes at specific locations across the North Atlantic, showing that oceanic carbon content has increased at a rate similar to (or greater than) the atmospheric increase from the mid 1980s to the present (e.g. Bates et al., 2014). The long time series sites illustrate that the anthropogenic trend is detectable over the interannual variability at their particular locations. The Atlantic interior carbon content has been studied by repeatedly revisiting hydrographic sections over several years to decades. These studies find significant decadal biogeochem-

ical variability associated with ocean circulation, and the anthropogenic increase is less obvious than at fixed observatories (Wanninkhof et al., 2010; Humphreys et al., 2016).

Numerical modelling studies also indicate that ocean circulation is the key driver of variability in the North Atlantic carbon sink. The spatial patterns of North Atlantic carbon content vary due to interannual and decadal atmospheric variability that modulates the exchange of heat, carbon, and alkalinity between the subpolar and subtropical gyres (Ullman et al., 2009; Thomas





et al., 2008; Doney et al., 2009; Tjiputra et al., 2012; Halloran et al., 2015). Furthermore, large interannual and decadal swings in the entrainment of surface water masses into the North Atlantic thermocline are responsible for the variability in the uptake of anthropogenic (Levine et al., 2011; Breeden and McKinley, 2016) and natural (DeVries et al., 2017) carbon. The total amount of carbon absorbed by the oceans by the end of the current century is sensitive to the total amount of anthropogenic

emissions (Ciais et al., 2013). At the same time, the inventory of natural carbon is expected to decrease by the year 2100 owing to reduced solubility and disequilibria, in spite of the accumulation of remineralised carbon (Bernardello et al., 2014).

Other work has investigated in detail carbon cycle variability over specific timescales, or in specific locations. In particular, other work has focused on the spatially varying patterns of change over time, yet relatively little attention has been given to the summed effect of these spatial patterns. This work aims to identify the key processes operating at the basin scale across a range

of time series, with the hope of improving our understanding of the expected behaviour of the North Atlantic over the coming century. We take the view that higher resolution models than this one are better suited to studying the system's functioning at finer scale. Similarly, coupled atmosphere-ocean simulations or ocean models forced with atmospheric reanalyses are more appropriate for the study of tightly coupled air-sea feedback phenonmena such as large scale modes of climate variability. High resolution and coupled atmosphere-ocean simulations are more costly to run and more complex to interpret, and so this work

seeks to provide insight that will be useful for subsequent study using more sophisticated tools.

Here, we quantify the variability of the dissolved inorganic carbon content of the North Atlantic basin over recent decades to the end of the 21$^{st}$ Century, and identify the mechanisms responsible. The main hypothesis is that oceanic uptake of anthropogenic carbon will dominate decadal and multidecadal changes in the North Atlantic carbon stock, while interannual variability will be driven by the solubility and soft tissue pumps. To address this hypothesis, a numerical model of the global

ocean is forced with output from a coupled climate model experiencing strong climate change due to 'business as usual' greenhouse gas emissions up to the end of the year 2099. First, a validation of modelled fields against observations of DIC and alkalinity is performed to establish that the simulations are appropriate to test the hypothesis. Next, a method is presented that partitions modelled fields of DIC into contributions from five components; saturation, soft tissue, carbonate, disequilibrium and anthropogenic. Then, modelled variability is separated into interannual, decadal and multidecadal timescales, and the long

term trend over 1860-2099. The role of each of the five components of DIC in controlling variability on each timescale is determined.

## 2 Methods

To explain variability in the carbon sink through time, a method of attributing DIC variability to underlying mechanisms is needed. To do so, the concepts of ocean carbon pumps (Brewer, 1978; Ito and Follows, 2005; Williams and Follows, 2011) are

applied in a computer simulation of the ocean and carbon cycle. A numerical model is a useful tool to disentangle the complex interplay between biogeochemical processes that control the ocean carbon sink. In a numerical model, it is possible to directly diagnose the roles of some of these processes, while the action of other mechanisms must be derived from model output. This section presents first a description of models used, then details of the configuration of simulations and finally the theoretical




framework used to partition the ocean carbon sink in terms of contributions from biogeochemical processes. Further details about these methods are provided in the Supplementary Information.

## 2.1 Model and Simulations

The ocean physical model used is the Nucleus for European Modelling of the Ocean (NEMO v3.2) (Madec, 2008), coupled
with MEDUSA-2 biogeochemistry (Yool et al., 2013a) and version 2 of the Louvain-la-Nueve Ice Model (LIM2, (Timmermann et al., 2005)) sea ice model. Simulations were configured with the ORCA1 grid, featuring a nominal horizontal resolution of 1-by-1 degree of latitude (about 110 by 110 km at mid-latitudes) with 64 depth levels (Madec and Imbard, 1996). The horizontal grid scale is finer near the equator (reaching approximately one-third by one-third degree of latitude) to better represent equatorial processes such as upwelling. MEDUSA-2 is an intermediate complexity biogeochemical model that resolves nitrogen,
silicon, iron, carbon, alkalinity and oxygen cycles. Further details about the models can be found in Couldrey et al. (2016) and the models' respective publications.

The ocean model was forced with output from a simulation from HadGEM2-ES, an earth system model (Collins et al., 2011). The atmospheric forcing covering the period 1860-2099 was taken from the HadGEM2-ES simulation, which was driven using prescribed greenhouse gas, land use, and atmospheric chemistry forcings according to Representative Concentration Pathway
8.5 (RCP8.5) (Jones et al., 2011) of the Intergovernmental Panel on Climate Change (IPCC). HadGEM2-ES features physical models of the ocean and atmosphere, the terrestrial and ocean carbon cycles, tropospheric chemistry and aerosols. The surface fluxes of heat, momentum and freshwater, and atmospheric chemistry from the HadGEM2-ES simulation were used to force NEMO at 6-hourly intervals. A 'control' forcing set was created by taking the first 30 years of this simulation (before changes in greenhouse gas concentrations impart meaningful climate change) and fixing greenhouse gas concentrations at a single
preindustrial value. The ocean model was then forced with repeats of the forcing set such that the model could be spun up for a much greater length of time than a single iteration of the HadGEM2-ES output would otherwise permit.

After 900 years of spin-up (see Appendix A), three simulations were spawned, each 240 years long, spanning 1860-2099 (inclusive). The Anthropogenic run (AN) features prescribed increasing atmospheric $CO_2$ (as well as other greenhouse gases: methane, nitrous oxide and halocarbons, land use change, and atmospheric aerosols) following RCP 8.5 (Jones et al., 2011),
which exceeds 900 ppmv (parts per million by volume, $11^{-1}$) by the end of year 2099 (Riahi et al., 2011). The Control (CN) simulation has its atmospheric $CO_2$ concentrations held fixed at a preindustrial level of 286 ppmv. The Control simulation was used to determine model drifts through the experiment. Another 'Warming Only' simulation was generated to distinguish the physical from the biogeochemical ocean carbon cycle response to rising atmospheric $CO_2$. This is achieved by decoupling the biogeochemical model from changing atmospheric $CO_2$ levels, while still allowing greenhouse gases to perturb the physical
model's radiation balance. In the Warming Only run, the model is subjected to the same radiative forcing of the Anthropogenic run, but atmospheric $CO_2$ is held constant at 286 ppmv. This allows for the ocean to physically adjust to anthropogenic climate change, without accumulating anthropogenic carbon. The carbon inventory changes in the Warming Only run are therefore conceptually similar to the 'non-steady state anthropogenic carbon' described by McNeil and Matear (2013).





## 2.2 Partitioning Reservoirs of DIC

The Anthropogenic simulation's DIC fields (C) are decomposed into contributions from saturation ($C^{sat}$), the soft tissue pump ($C^{soft}$), the carbonate pump ($C^{carb}$), disequilibria ($C^{dis}$), and anthropogenic emissions ($C^{anth}$), Eq. (1). The implementation of this partitioning framework is summarised here, and full details are included in the Appendix B. Other work describes how

the contributions of carbon pumps can be determined using commonly measured oceanographic parameters (Brewer, 1978; Ito and Follows, 2005; Williams and Follows, 2011). The approach taken in this work is conceptually similar, but takes advantage of model diagnostics rather than empirical estimation techniques.

$$C = C^{sat} + C^{soft} + C^{carb} + C^{dis} + C^{anth} \tag{1}$$

The saturation concentration of DIC ($C^{sat}$) is calculated using standard seawater carbon system equations using fields of

temperature, salinity, preformed alkalinity and a preindustrial mixing ratio of atmospheric $CO_2$ ($X_{CO_2}^{pre}$=286 ppmv) as input parameters. The DIC enrichment due to the soft tissue pump ($C^{soft}$) is calculated using fields of preformed and regenerated nutrient (generic Dissolved Inorganic Nitrogen or DIN, see Appendix A and (Yool et al., 2013a)) assuming that carbon is remineralised in a fixed ratio to nitrogen. The contribution of the carbonate pump ($C^{carb}$) is calculated using fields of 'regenerated' alkalinity, assuming that DIC is regenerated in a fixed ratio with alkalinity (see Appendix A for an explanation of 'preformed'

and 'regenerated' tracers). The anthropogenic carbon ($C^{anth}$) is calculated using the difference between DIC fields in the Anthropogenic and Warming Only simulations. Disequilibrium carbon ($C^{dis}$) is the remainder when the other four components are subtracted from the total DIC field. $CO_2$ disequilibria arise because water parcels are often subducted away from the ocean surface before they can equilibrate with the atmosphere. For example, if photosynthesizers assimilate $CO_2$ (during a strong bloom) more quickly than $CO_2$ can dissolve into seawater, and then that water parcel becomes subducted away from the surface

(by a storm), then the undersaturation will persist in the interior ocean as a negative $C^{dis}$ anomaly. Positive $C^{dis}$ anomalies may occur when a cold water parcel saturated with DIC is upwelled to the mixed layer and warms (which decreases $CO_2$ solubility) faster than $CO_2$ can outgas.

## 3 Model Validation

In this section, modelled DIC and alkalinity fields are compared against observations. GLODAPv2 (Key et al., 2015; Olsen

et al., 2016) is a merged, internally consistent, global data set of seawater biogeochemical parameters including DIC and alkalinity, as well accompanying physical water properties dating between 1972 and 2013. Since the ocean-only model is forced with output from an atmospheric model rather than observations, variability in the model will not be correlated with real world variability. In other words, at any grid cell for a single point in time, the modelled and observed monthly mean DIC, alkalinity etc. will differ because the model experiences environmental variability that is not correlated with real world

variability at that time. As such, it is more meaningful to compare the statistical properties of modelled variability, than to carry out a one-to-one comparison with observations.



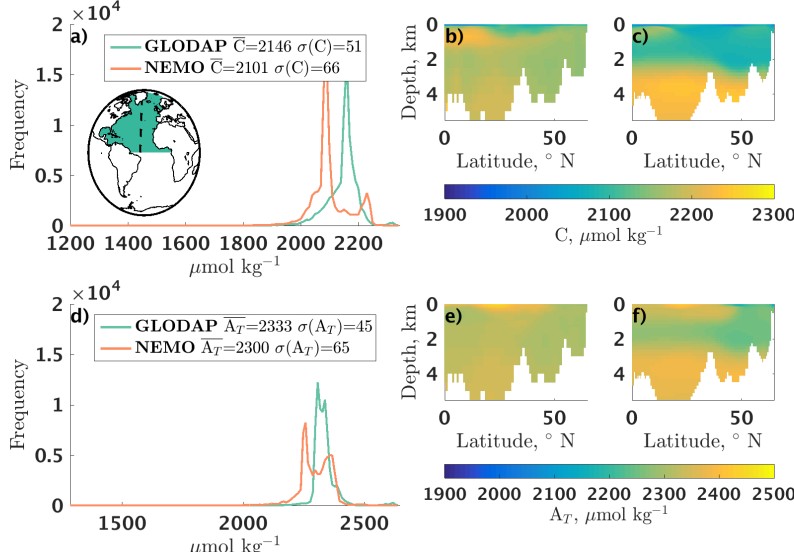

**Figure 1.** Frequency distributions of North Atlantic DIC (a) and alkalinity (d) in the unmapped GLODAPv2 database (green) with their corresponding values from colocated model grid cells from the NEMO-MEDUSA Anthropogenic simulation (orange), where horizontal axes limits reflect the ranges of values. Time and space means of all DIC and alkalinity values are indicated in the legends. Depth transects of DIC along 35.5° W for the observations (b) and model (c). Depth transects of alkalinity along 35.5° W for the observations (e) and model (f). Green area on inset map in (a) indicates the North Atlantic domain of this study, and the black dashed line indicates the location of the section along 35.5° W

To create a collection of model output comparable to the observations in GLODAPv2, the model grid cell values of DIC and alkalinity nearest (in space and time) to each unmapped GLODAPv2 data point were identified. These two collections of modelled and observed DIC and alkalinity are then compared for their spatial and temporal variability by comparing the aggregrated frequency distributions of all values of DIC and alkalinity in 10 $\mu$mol kg$^{-1}$ bins (Fig. 1a and d). This demonstrates
5   the extent to which the modelled fields of DIC and alkalinity show realistic spatial and temporal variance, in spite of the paucity of observations and differing histories of variability. By comparing model grid cells with unmapped observations, a more direct validation of the carbon system is made than if an interpolated or mapped observation set were used. Since this work focuses on the North Atlantic, only datapoints from this basin are compared to model output; this domain is shown in the inset map of Fig. 1a.
10   The modelled fields of DIC and alkalinity show broad agreement but with clear statistical differences from the data (Fig. 1). Modelled DIC shows a low mean bias of 2101 $\mu$mol kg$^{-1}$ versus the 2146 $\mu$mol kg$^{-1}$ observed and, spatiotemporal variability is somewhat overestimated (66 $\mu$mol kg$^{-1}$ versus the 51 $\mu$mol kg$^{-1}$ observed). The most discrepant DIC values (the secondary mode with values above 2200 $\mu$mol kg$^{-1}$ in Fig. 1a) are associated with a high bias in the model below 3000 m. Modelled mean



North Atlantic alkalinity shows a low bias (of 33 $\mu$mol kg$^{-1}$) and excessive spatiotemporal variability (standard deviation of 65 $\mu$mol kg$^{-1}$ versus 45 $\mu$mol kg$^{-1}$).

The time mean spatial structure of modelled DIC and alkalinity is assessed against the GLODAPv2 mapped climatology (Lauvset et al., 2016). Key features of the North Atlantic DIC vertical structure (Fig. 1b) are represented by the model (Fig. 1c);

low values in the surface 100 m that increase with depth, and a prominent enrichment around 750 m at low latitudes. The the bimodal distribution of DIC in the model (Fig. 1a, orange line) results from the overestimation of the surface to depth contrast of DIC the model. As with DIC, the model (Fig. 1f) also captures some of the broad structure of the North Atlantic distribution of alklainity (Fig. 1e). The model represents the shallow (0–1 km) alkalinity maximum in the mid latitudes well, but elsewhere north–south and surface–depth contrasts are overestimated. The deep (>2 km) ocean tends to have little impact on interannual

to multidecadal variability at the basin scale, and model errors in the background mean state are deemed unimportant for this study.

The biological carbon pump is crucial mechanism for the North Atlantic carbon sink, yet it is poorly understood and therefore difficult to validate in models. A recent data-based synthesis estimated the time mean strength of the modern North Atlantic biological carbon pump to be 0.55-1.94 PgC yr$^{-1}$ (Sanders et al., 2014). Modelled export production averaged over the period

1990-2009 in this simulation is 0.78 PgC yr$^{-1}$ (i.e. within the observed range) with annual mean values for the basin varying between 0.75 and 0.82 PgC yr$^{-1}$ over the period. The strength of the simulated North Atlantic biological carbon pump is therefore realistic.

The model's Total Primary Production (TPP) is a key component of the ocean carbon cycle, and is assessed here. The strength of global primary production in the model (averaged over 2000-2004, inclusive) is 51.1 PgC yr$^{-1}$, which is on the

low end of a range of estimates based on independent empirical models over a similar period: 58.8, 60.4, and 46.3 PgC yr$^{-1}$ (Behrenfeld and Falkowski, 1997; Carr et al., 2006; Westberry et al., 2008). Over the same time period, annual mean modelled TPP varies between 50.6 and 51.5 PgC yr$^{-1}$. Specific to the North Atlantic, MEDUSA-2 underestimates TPP strength during the subpolar spring bloom, as well as annual mean TPP at lower latitudes (Yool et al., 2013a) (their Fig. 12). Robust primary productivity fields in the model are only important insofar as they produce a reasonable export of sinking particulate matter.

Since modelled export production in the North Atlantic is within the estimated range for the real world, the biases in the region's primary production are small enough to be acceptable. A more broad-scope assessment of primary production in MEDUSA-2 is discussed by Yool et al. (2013a), their Section 4.1. Based on this comparison and the validation work done by Yool et al. (2013a), although biases certainly exist, the carbon cycle in the model can be judged to be reasonable and our setup is fit for this study.

## 30  4   Results

### 4.1   Description of 21$^{st}$ Century North Atlantic DIC variability

This section reports the magnitude of the North Atlantic DIC inventory variability, and then attributes components of this variability to underlying processes. Before quantitatively attributing DIC variability to underlying causal mechanisms, the



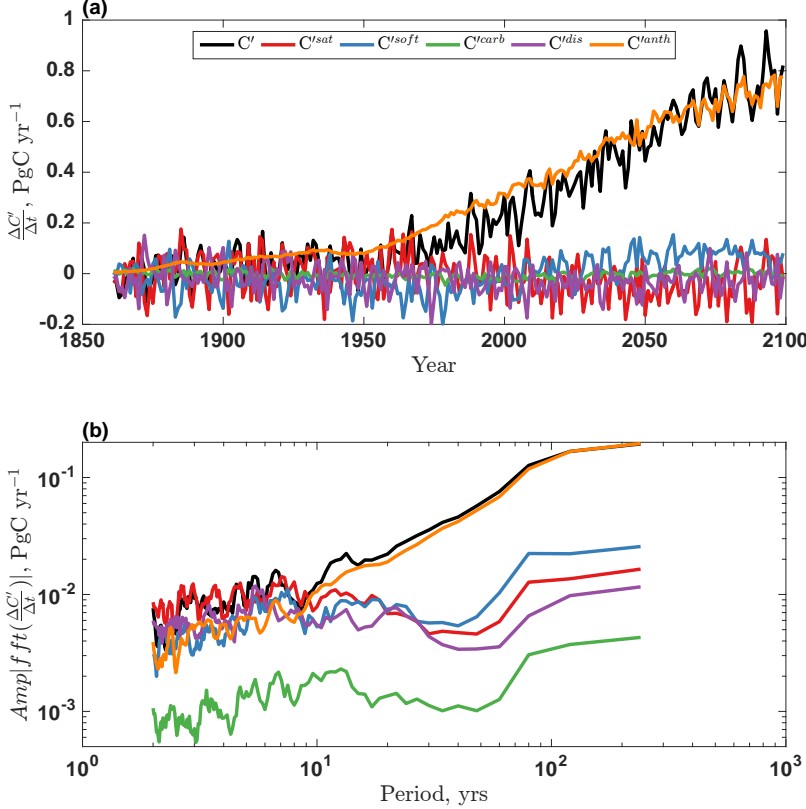

**Figure 2.** a) First time derivative of total modelled North Atlantic carbon reservoir anomalies, black, with DIC component reservoirs, color. b) Amplitudes of spectra of modelled variability of the inputs shown in (a) against period in years. Note both logarithmic axes, and spectra are shown with short periods (high frequency) on the left, increasing to long periods (low frequency) on the right

spectrum of the North Atlantic's carbon inventory is shown. First, the North Atlantic's full volume integrated inventory of DIC is calculated each year using annual mean fields (the North Atlantic domain is the same as in Fig. 1a). The same is then done for the components of DIC: $C^{sat}$, $C^{soft}$, $C^{carb}$, $C^{dis}$ and $C^{anth}$. Anomalies of these inventories (denoted with a prime, $C'$) are calculated by subtracting out the long term (1860-2099) mean from each component. To first show the time variability of DIC and components, period spectra are calculated using standard Fast Fourier Transform techniques. It was not possible to satisfactorily remove the time trend from the series of $C'$ and its components (as one ideally would) before calculating spectra since their trends were not always well described by linear functions. Instead, more meaningful spectra are found when calculated from the time derivative (as a finite time difference) $\frac{\Delta C'}{\Delta t}$ (Fig. 2a). This detrending method is discussed in Appendix G. Spectra are smoothed with a 5-point moving average to aid visual comparison (Fig. 2b).



Comparing the variability of the total carbon reservoir, C′, with its components in Fig. 2 begins to show which mechanisms are reponsible for which variability. The long term accumulation of C′ closely matches that of C′$^{anth}$ (black and orange lines respectively in Fig. 2a and b), especially for periods longer than 10 years. There is substantial variability in C′ on shorter timescales. The close alignment of the spectra for C′ and C′$^{sat}$ in the 1-10 year period band suggests that the C′$^{sat}$ component

is important on these shorter timescales. Across all timescales, the spectrum for C′$^{carb}$ shows low amplitudes, suggesting this component is minor in determining the variability of the North Atlantic carbon sink. The other two components, C′$^{soft}$ and C′$^{dis}$ show moderate to large amplitudes across a range of timescales. However, without an additional understanding of how component variability correlates with total DIC variability, it is not possible to discern the crucial detail of the interplay between components. This analysis reveals that North Atlantic DIC variability consists of different bands of timescales, each controlled

by a different set of mechanisms.

$$C'^{Trend} = \sum_{i=0}^{3} a_i t^i \tag{2}$$

$$C'^{Detrend} = C' - C'^{Trend} \tag{3}$$

$$C'^{Multidecadal} = f_{20}(C'^{Detrend}) \tag{4}$$

$$C'^{Decadal} = f_5(C'^{Detrend}) - C'^{Multidecadal} \tag{5}$$

$$C'^{Interannual}3 = C'^{Detrend} - C'^{Multidecadal} - C'^{Decadal} \tag{6}$$

Next, four distinct timescales of variability of the total North Atlantic DIC inventory (Fig. 3) are isolated and examined. The long-term response of the North Atlantic carbon sink (or trend) is quantified by fitting a third order polynomial to annual mean DIC inventory anomalies (C′) from a long term mean, Eq. (2), where $a_0, ..., a_3$ are polynomial constants and $t$ is time (Fig. 3e). This long term trend is subtracted from the C′ time series to quantify variability on other timescale, Eq. (3). A discussion of the

different detrending methods between this analysis and the earlier Fourier analysis is included in Appendix G. Low frequency multidecadal variability (without the trend) with ∼0.75 PgC of peak-to-peak variability is revealed by applying a low-pass 20 year running mean ($f_{20}$) to the detrended C′ series, C′$^{Detrend}$, Eq. (4) (Fig. 3d). The medium frequency decadal (5-20 year) variability is calculated by filtering the detrended series with a 5 year running mean ($f_5$), then subtracting the low frequency series, Eq. (5) (Fig. 3c). Finally, the high frequency interannual (1-5 year) variability is the detrended series with the low and

medium frequency components subtracted out, Eq. (6), and is on the order of one tenth of one PgC (Fig. 3b). Note that the seasonal cycle is not included in this analysis, since annually averaged model output is used. This same filtering analysis is applied to the time series of each of the components of the total DIC reservoir: saturation, soft tissue, carbonate, disequilibrium and anthropogenic. These four timescales of DIC variability were chosen because each is controlled by a different set of mechanisms.

Examining the spatial structure of the variability of column integrated DIC in the North Atlantic provides an indication of which processes may control the inventory at the basin scale. At each location, the column inventories of annually averaged DIC fields are calculated by summing through depth at each horizontal location. Next, the root-mean-square (RMS, see Appendix





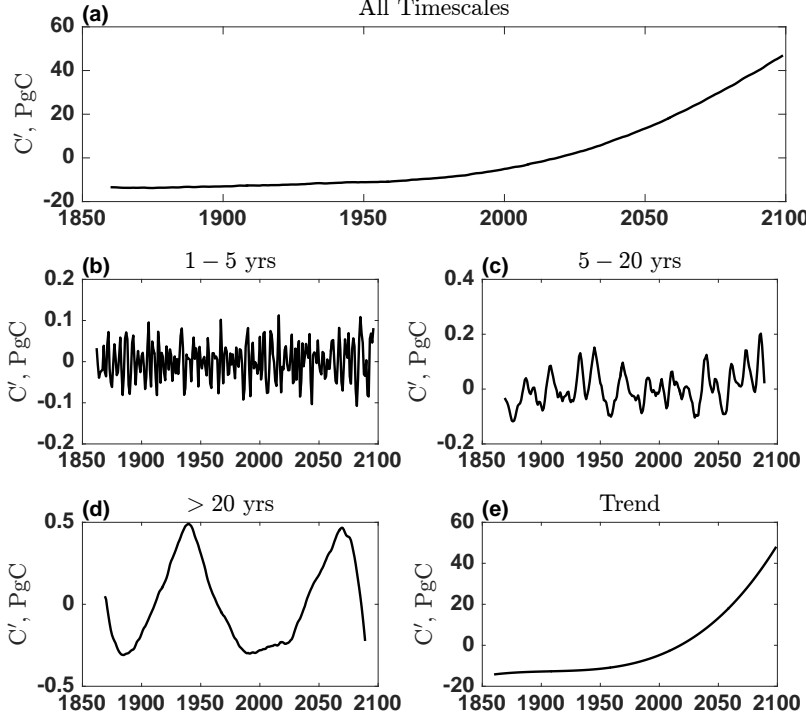

**Figure 3.** Variability of the North Atlantic Inventory shown as the root-mean-square of DIC anomalies from a long term mean (a), showing the interannual (b), decadal (c) and multidecadal (d) components, and the long term trend (e)

C) of depth-integrated DIC anomalies is calculated, which shows where carbon variability is large, Fig. 4. The main sites of interannual DIC variability (RMS > 20 $gCm^{-2}$) are in the transition regions at the northern and southern edges of the subtropical gyre, and within the subpolar gyre, Fig. 4a. Large decadal variability (RMS > 20 $gCm^{-2}$) is found in the western subpolar gyre, along the path of the North Atlantic Current, and at the subtropical/tropical interface, Fig. 4b. The basin's

5   strongest multidecadal variability (RMS > 40 $gCm^{-2}$) is concentrated in the subpolar zone and high latitude seas, Fig. 4c. Across the entire North Atlantic, there are large long-term column inventory carbon anomalies (RMS > 500 $gCm^{-2}$), especially along the western boundary, Fig. 4d. While images of the basin at the grid point-scale show where variability is hosted, it is also necessary to aggregate the entire domain's inventory to understand the basin at the larger scale. Figure 4 does not reveal how patterns of variability at different locations compensate each other, and so a basin-integrated perspective is also needed.

10     The maps of column integrated DIC variability pose several hypotheses about which processes might dominate the variability of the entire basin's inventory. 1) Large interannual DIC variability at the gyre edges suggests that physical redistribution of DIC within the basin is important. This process cannot fully explain variability in the basin's total inventory since redistributing DIC around the basin will produce local anomalies that together sum to zero, and so can be immediately discounted. 2) Large





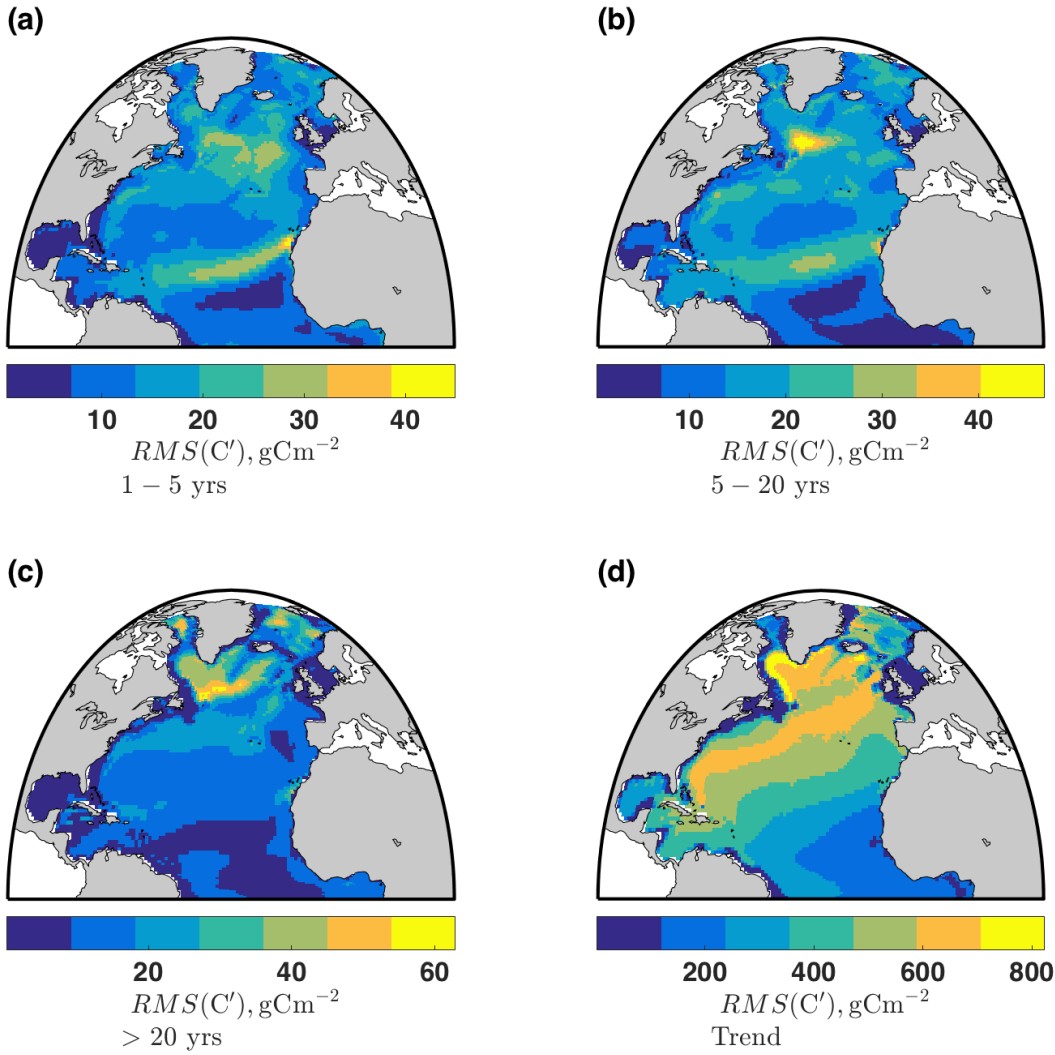

**Figure 4.** Variability of the North Atlantic Depth-Integrated Inventory of DIC anomalies from a long term mean, showing the 1–5 year interannual (a), 5–20 year decadal (b) and > 20 year multidecadal (c) , and the long term trend (d) components

interannual DIC variability in the subpolar gyre could be the result of strong biological variability modulating a) biological carbon fixation, creating DIC disequilibria and/or b) export production of soft tissue carbon. 3) Physical variations in high latitude deepwater formation affect the saturation concentration of DIC by varying the subduction of heat and alkalinity into the deep ocean. 4) Wind variability along the storm track through the subpolar gyre causes large variance in the gas transfer velocity, which varies the strength of $CO_2$ fluxes, and in turn modulates the subduction of DIC disequilibria. 5) Ice cover




variability at high latitudes modifies DIC saturation and disequilibria by insulating seawater from the atmosphere in terms of heat and gas exchange. 6) Very large anomalies due to the long term trend along the western side of the basin result from anthropogenic $CO_2$ being carried southward by North Atlantic deep waters. 7) North Atlantic DIC inventory size is controlled by advection into or out of the basin, rather than processes occurring within the basin. The next sections of analysis will first

identify which components of DIC control the integrated basin inventory variability across different time scales, and then attribute that variability to oceanographic processes to narrow down this list of hypotheses.

### 4.2 Attribution of Basin-Integrated DIC Variability to Causal Mechanisms

The next part of the analysis details a stastical description of the entire North Atlantic carbon inventory variability. To asses the magnitude of variability on each timescale, the RMS is calculated for each of the basin-integrated, temporally filtered DIC

components (Fig. 5a, d, g and j). This shows how large the anomalies in each DIC component are, with respect to the total DIC anomaly. However, for a component to be a strong control on the total DIC variability, the anomalies in that component must be both large and well-correlated with DIC anomalies. For example, although decadal variability in $C^{soft}$ is quite large (RMS = 0.07 PgC, Fig. 5d), in comparison to C′ (RMS = 0.06 PgC), anomalies in $C^{soft}$ are not correlated with C′ (correlation r < 0.1, Fig. 5e). Therefore, $C^{soft}$ is not a strong control on North Atlantic decadal DIC variability. Equally, if component

anomalies are well-correlated with C′, but small with respect to C′ (as in the case of $C^{sat}$ for the long term trend, Fig. 5j and k) then its overall impact on C′ is weak. A useful metric to quantify the control of a mechanism on DIC variability is the gradient of the regression line drawn between filtered carbon anomalies (C′) and anomalies in each component (Fig. 5c, f, i, l). This dimensionless number has values close to 1 if a component strongly controls C′, and close to 0 if the component is unimportant. Values for the gradient of one component may be larger than 1 if at least one other component's gradient is

negative (i.e. compensatory). If the relationship between each component and the total DIC anomaly is perfectly linear then the sum of the gradients of all components is equal to one. However, in practice these relationships are not precisely linear, but nevertheless the sum of gradients is always close to 1 (0.89–1.14). By examining all three of these statistics (RMS of anomalies, correlations and regression gradients), one can attribute North Atlantic DIC inventory variability to its constituent components.

A comparison of the size of variability across the four different timescales as the RMS of anomalies (Fig. 5a, d, g, j) shows that the magnitude of DIC inventory variability increases for longer timescales. In particular, interannual anomalies (Fig. 5a) are much smaller (on the order $10^{-2}$ PgC) than longer timescale anomalies (order $10^{-1}$ to $10^1$ PgC, Fig. 5g, j). While it would be tempting to dismiss these interannual anomalies as unimportant since they are small in comparison with the long term changes in basin carbon content by the year 2100, they are nonetheless crucial in setting the interannual variability of

the ocean's uptake of $CO_2$, which in turn is an important component of climate variability and predictability (McKinley et al., 2017).

Interannual North Atlantic DIC variability is attributed primarily to the saturation component, $C^{sat}$ (regression gradient of 0.78). $C^{anth}$ and $C^{carb}$ are minor contributors on this timescale (0.10 and 0.05), and the roles of $C^{soft}$ and $C^{dis}$ are negligible (Fig. 5c). Note that while a carbon anomaly in a given year may have large components from various periods of variability, this





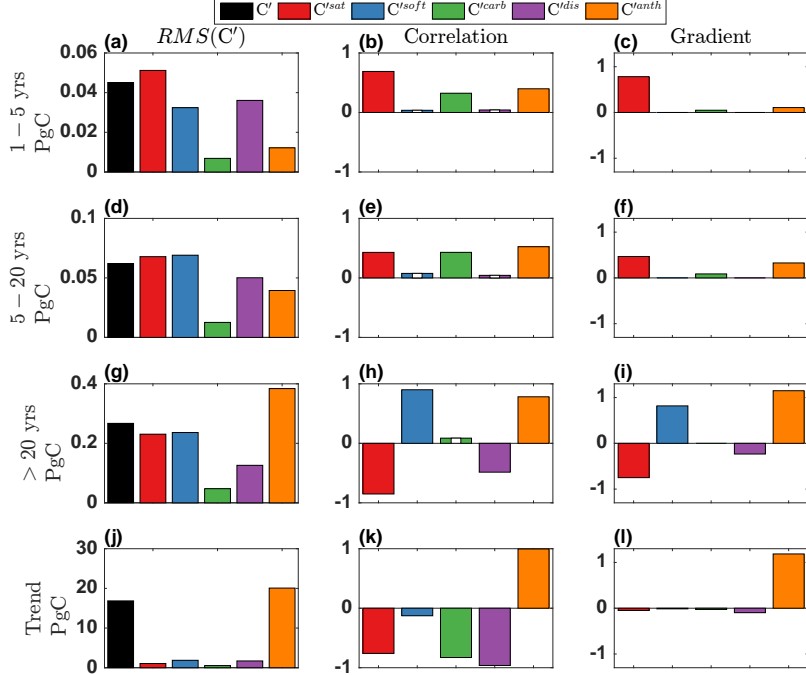

**Figure 5.** Statistical data treatment describing the relationship between DIC component variability and total DIC inventory variability on four timescales: interannual (1–5 year, a-c), decadal (5–20 year, d-f), multidecadal (>20 years, g–i), and the long term trend (j–l). Left: root-mean-square (RMS) of anomalies of total DIC inventory (black) and DIC components (colors). Center: Correlations between component and total DIC inventory anomalies. Correlations insignificant at 95% confidence (p value > 0.05) have a white bar through the middle. Right: gradient of regression line between component and total DIC inventory anomalies (insignificant gradients with p > 0.05 not plotted).

analysis identifies the portion of that anomaly that varies on specific timescales. For example, for a given time, a large part of the total carbon anomaly C′ may be due to a large anomaly in C$^{soft}$, but if that C$^{soft}$ anomaly is part of long period variability, then this analysis will find C$^{soft}$ to be unimportant on interannual timescales and instead identifies which component varies coherently on short, interannual timescales.

5    On decadal timescales, most DIC variability is driven by the C$^{sat}$ (regression gradient of 0.46) and C$^{anth}$ (0.32) components, each with similar importance (Fig. 5f). Over multidecadal timescales, large variability in C$^{sat}$ (-0.76) and C$^{soft}$ (+0.81) is approximately compensatory, and variability in the accumulation of anthropogenic carbon (1.16) dominates the variability of the North Atlantic carbon sink. The longest-term changes in the North Atlantic DIC inventory are driven almost entirely (1.18) by the basin's uptake of anthropogenic carbon (Fig. 5l). On this timescale, although there are considerable changes in the

10    inventories of each of the other components on the order of a few PgC, these are dwarfed by the basin's accumulation of 66 Pg of anthropogenic carbon (Fig. 3e). Over the course of the Anthropogenic simulation (after accounting for model drift, see




Appendix E), saturation carbon declines by 2.12 Pg, soft tissue carbon increases by 1.56 Pg, carbonate pump DIC decreases by 1.50 Pg, disequilibrium carbon decreases by 2.73 Pg.

## 4.3 Roles of Temperature versus Alkalinity Effects on $C^{sat}$ Variability

The previous section has established that variability in saturation carbon is an important driver of the variability of total North
Atlantic DIC in this model, and so the following section seeks to account for this. Using the same variability analysis as for the total DIC inventory, the sources of variability in $C^{sat}$ are quantified. For this analysis, only 3 timescales of variability are distinguished, since the controls on decadal and multidecadal $C^{sat}$ variability are similar. To investigate the contribution of temperature, alkalinity and salinity to variability in $C^{sat}$, fields of $C^{sat}$ are calculated offline (using model output rather than at each time step during the simulation) by fixing two fields while allowing the other to vary. For example, to quantify the
$C^{sat}$ variability caused by variability in ocean temperature, $C^{sat}_{temp}$, fields of $C^{sat}$ are calculated using annual mean temperature fields for each year, but the alkalinity and salinity fields from the first year of the experiment, 1860. Note that this method of calculating saturation concentration by varying one parameter and fixing the others does not account for nonlinearities in carbon system equations. This means that at a given time, the sum of $C^{sat}$ anomalies due to temperature, preformed alkalinity and salinity will not exactly equal the $C^{sat}$ anomaly when all three parameters vary together. Nevertheless, this method of
decomposing $C^{sat}$ variability is useful because the errors that result from carbonate system nonlinearities are much smaller than the effects of interest.

Across all timescales, temperature and alkalinity are both important seawater properties that control $C^{sat}$ variability, and the impact of salinity is negligible. On interannual timescales, large anomalies in $C^{sat}$ due to temperature (regression gradient of 0.60) and alkalinity (0.49) variability (Fig. 6b) are well-correlated with the total $C^{sat}$ variability (Fig. 6c), and so both play
roles in controlling interannual $C^{sat}$ variability (Fig. 6d). On decadal and multidecadal timescales, variability in temperature imparts most of the variability in $C^{sat}$ (0.71), but alkalinity variability is also important (0.37, Fig. 6g). The trend in $C^{sat}$ arises due to the partially compensatory long term changes in North Atlantic temperature and alkalinity (2.58 versus -1.65), which act in opposition to each other (Fig. 6i and j). The effect of the warming climate on ocean temperature, however, is larger by a factor of 1.5, and gives rise to the long term decline of $C^{sat}$ in the North Atlantic (Fig. 6a). The only timescale over which
salinity variability has an impact on $C^{sat}$ is in the long term, causing a very small decrease by the end of 2099 (Fig. 6j).

## 4.4 Role of $C^{sat}$ in Interannual DIC Variability

Having established that $C^{sat}$ strongly dominates interannual variability in the total North Atlantic DIC inventory, this section investigates how closely one may estimate interannual DIC variability using only seawater temperature and salinity. Thus far, $C^{sat}$ has been calculated in the model using fields of seawater temperature and preformed alkalinity, where the latter is
explicitly modelled as a tracer. In the real world, no such tracer exists, but preformed alkalinity can be estimated by assuming that both it and salinity are mixed conservatively in the ocean (see Appendix D). This method of estimating $A_T^{pre}$ (and hence, $C^{sat}$) is not as robust as through the use of explicitly modelled tracers, because the relationship between surface salinity and





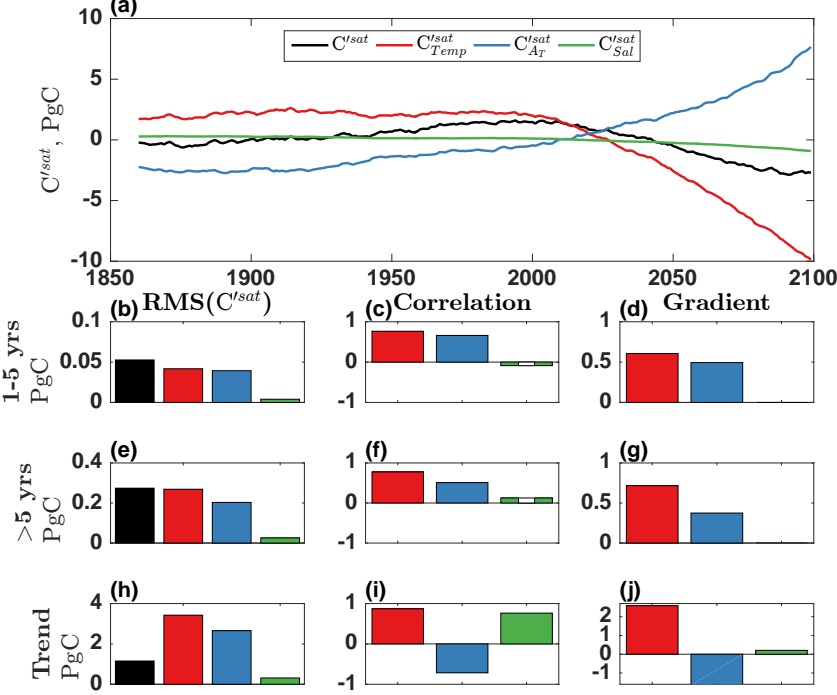

**Figure 6.** Time series of $C'^{sat}$ with contributions from temperature, alkalinity and salinity (a), with descriptive statistics showing the roles of each contribution on different timescales, : interannual (1-5 years, b–d), decadal and multidecadal (>5 years, e–g), and the long term trend (h–j). Left: root-mean-square (RMS) of anomalies of $C^{sat}$ inventory (black) and components (colors). Center: Correlations between component and total $C^{sat}$ inventory anomalies. Correlations insignificant at 95% confidence (p value > 0.05) have a white bar through the middle. Right: Gradient of regression line between component and total $C^{sat}$ inventory anomalies (insignificant gradients with p > 0.05 not plotted)

alkalinity is not precisely linear. However, it is useful to examine here, because $C^{sat}$ estimated using an empirical relationship for $A_T^{pre}$ is easily measureable in the real ocean.

First, the model's North Atlantic basin integrated annual mean inventory of DIC is calculated over a forty year period (1980-2019), which represents the contemporary ocean. The anomalies of the DIC inventory ($C'$) from the 1980-2019 mean are then calculated (Fig. 7a, black line). Over the same period, anomalies of the basin's saturation DIC inventory are calculated, using fields of preformed alkalinity that have been parameterised from salinity (Fig. 7a, red line). The anomaly inventories $C'$ and $C'^{sat}$ are then detrended (using a quadratic function for each), then high-pass filtered with a 5 year window to select only the interannual variability (i.e. all variability with periods longer than 5 years are removed), Fig. 7b. Visual comparison of the





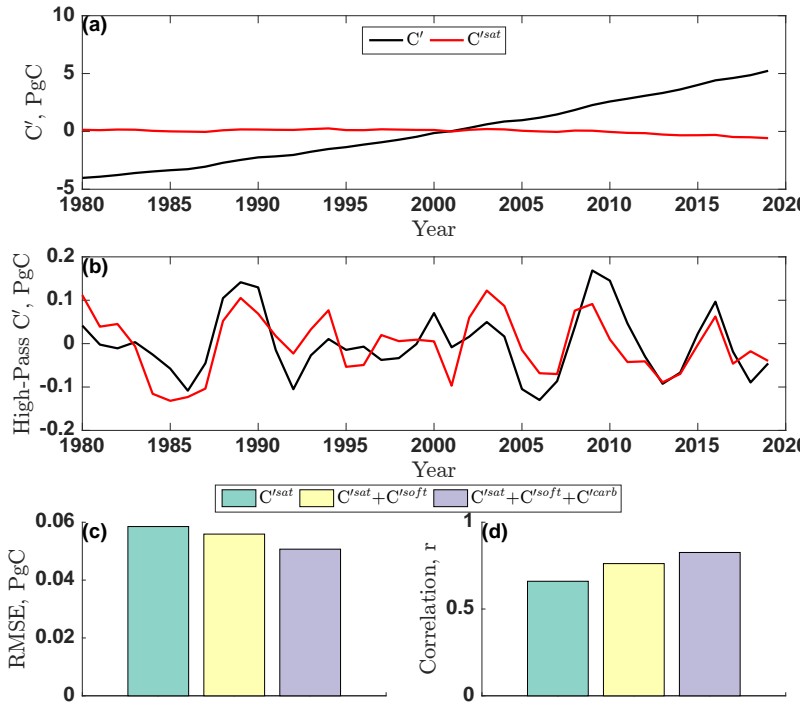

**Figure 7.** Comparison of total North Atlantic DIC inventory variability and $C^{sat}$ over 1980–2020, showing (a) anomalies of total DIC ($C'$, black) and $C^{sat}$ (red) from a long term mean. (b) Detrended and high-pass filtered anomalies (periods of 5 years and shorter) showing interannual variability of $C'$ (black) and $C^{sat}$ (red). (c) Root-Mean-Square Error (RMSE) of high-pass filtered DIC component combinations from high-pass filtered $C'$. (d) Correlation between combinations of high-pass filtered DIC components and high-pass filtered $C'$ (all p values $< 0.05$)

interannual anomalies in $C'$ and $C'^{sat}$ shows that the two agree to first order. The Root-Mean-Square-Error of $C'^{sat}$ from $C'$ is 0.058 PgC and the two are moderately correlated (r = 0.66, p < 0.05).

Clearly, there is a certain amount of interannual variability in $C'$ that cannot be estimated from $C'^{sat}$ alone, and the difference comes from the other components of DIC. The soft tissue carbon component ($C^{soft}$) can be parameterised using Apparent
5  Oxygen Utilization (see Appendix D), and then interannual basin-integrated anomalies, can be calculated by detrending and high-pass filtering in the same way as for $C^{sat}$. One may quantify how adding this additional component of DIC to the $C'^{sat}$ anomalies improves the estimate of the total basin variability. The agreement between filtered $C'$ and $C'^{sat}+C'^{soft}$ (RMSE = 0.055 PgC, r = 0.76, p < 0.05) is improved somewhat compared to $C'^{sat}$ alone. Adding in the component of DIC from the carbonate pump (calculated using preformed alkalinity that has been parameterised using salinity, see Appendix D), improves
10  the resemblance to $C'$ further still (RMSE = 0.050 PgC, r = 0.82, p < 0.05), but not by a great deal. The remainder of the

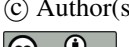



misfit between the combination of components and C′ comes from a mixture of C′$^{dis}$, C′$^{anth}$ and errors that result from parameterising preformed alkalinity and C$^{soft}$, which require detailed observations of the carbon system to constrain (any pair of DIC, A$_T$, pH and $p$CO$_2$). The result is that a first order approximation of interannual North Atlantic DIC variability can be readily estimated using observations of North Atlantic temperature and salinity. Further, including the C$^{soft}$ component

(which can be estimated using oxygen measurements) improves the estimate of DIC variability.

## 5 Discussion

In this work, it was shown that different mechanisms give rise to variability in the size of the North Atlantic carbon sink over a range of timescales through the industrial era to the end of the current century. In general, variability primarily in temperature but also in preformed alkalinity of the North Atlantic governs the interannual variations in basin DIC content.

Variance in saturation carbon is a factor of 1.8 times stronger than the sum of variances of all the other components. This result suggests that relatively high frequency variations in ocean heat content and preformed alkalinity are critical in driving short term changes in the carbon reservoir, which are likely to govern short term CO$_2$ flux variability (Halloran et al., 2015). Previous work describes how the physical redistribution of heat and DIC in the upper 100 m is the main process giving rise to interannual variability in surface ocean $p$CO$_2$ and therefore air-sea carbon fluxes in the North Atlantic (Doney et al., 2009;

Ullman et al., 2009; Tjiputra et al., 2012). However, local anomalies due to advection can compensate each other and sum to zero change in the basin carbon inventory, for example if DIC simply moves from one location to another. Advection can stimulate total inventory changes if, for example, DIC-rich thermocline water becomes entrained into the mixed layer and outgasses. The basin-integrated approach to carbon inventory analysis has the advantage that compensating anomalies cancel out, and only processes that yield net inventory change are revealed.

Further work would be necessary to exhaustively identify the roles of all mechanisms that vary temperature and preformed alkalinity (such as surface heat and freshwater fluxes, biological modulation of surface alkalinity etc.) Nevertheless, some potential mechanisms could be readily checked. Correlation between interannual variability in the strength of the Atlantic Meridional Overturning Circulation (AMOC) and the basin-integrated DIC inventory was very low (r < 0.1, see Appendix F). This low correlation indicates that the AMOC is not the main process driving interannual variability in North Atlantic carbon

content (although it is clearly important on the longer timescales shown here and much longer glacial-interglacial timescales (Hain et al., 2014)). The basin-integrated DIC inventory variability is moderately correlated (r= 0.43) with the mean DIC concentration at the open southern boundary of the basin (the equator), suggesting that the interannual DIC variability described here mostly arises within the basin, but partly reflects variability at the open southern boundary (Appendix F). Finally, the total DIC inventory variability is weakly correlated (r< 0.2) with the basin mean temperature, so interannual DIC variability cannot

be explained purely in terms of ocean heat content modifying solubility.

Interannual DIC variations are small in size in comparison with the longer-period variability of North Atlantic carbon content. The accumulation of anthropogenic carbon on decadal and multidecadal timescales becomes increasingly important with time in setting the basin's DIC content. Even so, saturation carbon variability remains an important driver of total DIC





variability across decadal to multidecadal timescales. On multidecadal timescales, fluctuations in the basin's soft tissue carbon reservoir also become critical. These soft tissue reservoir variations are almost completely counteracted by variations in the basin's saturation carbon content of opposing sign, with the net effect on total DIC being primarily driven by anthropogenic carbon accumulation. Again, this saturation carbon variability is mostly temperature driven (which increases in response to

anthropogenically forced climate change), with preformed alkalinity playing a secondary role. Further work should investigate the relative importance of the production of $C^{soft}$ by biological processes versus physical horizontal and vertical transports. Bernardello et al. (2014) describe how the natural carbon cycle response at the end of the 21$^{st}$ Century is characterised by large, nearly compensating decreases in preformed carbon (due to decreased $C^{sat}$) and increases in $C^{soft}$. The work presented here identifies the same long-term DIC response (the trends in $C^{sat}$ and $C^{soft}$, Fig. 5l), but also that counteracting $C^{sat}$ and

$C^{soft}$ variability is a critical control on North Atlantic DIC on shorter, multidecadal timescales as well (Fig. 5i).

Large multidecadal North Atlantic DIC variability appears to be connected to the basin's circulation. Sanchez-Franks and Zhang (2015) describe how decadal and longer timescale fluctuations in AMOC strength are correlated with the nutrient supply to the photic zone in the North Atlantic, which in turn modulates primary productivity. If export production is tightly coupled to primary productivity, then one might also expect that decadal and multidecadal AMOC variability in turn controls the creation

of $C^{soft}$. However, the overturning has additional physical effects on $C^{soft}$: stronger overturning could also be associated with an increased removal of $C^{soft}$ from the basin, either by outward horizontal advection or upwelling into the mixed layer (where $C^{soft}$ is converted into $C^{dis}$, see Appendix B). Indeed, redistribution of $C^{soft}$ has been suggested as the main driver of observed multidecadal DIC accumulation in the northeast subpolar Atlantic (Humphreys et al., 2016). Furthermore, the role of the upper (0-1000 m) Atlantic overturning in controlling decadal $CO_2$ flux variability by modulating the supply of remineralised DIC

($C^{soft}$ and $C^{carb}$) has been discussed recently (DeVries et al., 2017). Those authors link decades of stronger overturning in the upper 1000 m with increased outgassing of natural (non-anthropogenic) $CO_2$. Future work should aim to distinguish the roles of creation, advection, and destruction of soft tissue carbon, and to identify the roles of specific advective pathways in and out of the North Atlantic. Similarly, the North Atlantic anthropogenic carbon content shows large multidecadal swings in phase with the total DIC. The AMOC is a likely candidate responsible for these multidecadal swings, given the strong correlation

(r$= -0.78$) between the two (see Appendix Fig. F1) although further work is necessary to explicitly link the mechanisms.

The model validation highlights that although the North Atlantic carbon and alkalinity fields show reasonable agreement with the observations at the basin scale, there are differences in local details. Therefore, the model is an appropriate tool to study the basin (rather than gridpoint or local) scale. The validity of this work relies on the balance between modelled physical, chemical, and biological controls on DIC variability being realistic. The large scale estimates of $CO_2$ flux variability discussed

by Couldrey et al. (2016) (who use very similar simulations) and the primary production and export production values described in Section 3 indicate that the model agrees well with available observations. Rather than provide definitive answers, these model results therefore represent a useful mechanistic insight into the North Atlantic carbon cycle that should serve to guide further study with more sophisticated tools (for example, coupled atmosphere-ocean models or Earth System Models).



## 6   Conclusions

Variability in the North Atlantic carbon sink across interannual, decadal, multidecadal and centennial timescales was attributed to contributions from driver processes. In this model, interannual anomalies of DIC in the North Atlantic are mainly driven by temperature and alkalinity variations to the saturation concentration. Most of basin's decadal variability is set by saturation variability, and anthropogenic carbon plays an important secondary role. On multidecadal timescales, swings in large scale North Atlantic circulation drive strong compensating effects on DIC: decreasing saturation and increasing soft tissue carbon variability mostly cancel each other, and the anthropogenic increase drives the net effect. The long term trend is entirely characterised by the North Atlantic's accumulation of 66 PgC$^{anth}$, which dominates the comparatively minor changes of a couple of PgC in the natural pools by the end of the 21$^{st}$ Century.

The atmosphere is rapidly accumulating $CO_2$ due to human activities, with potentially large impacts on the climate system and human society, however the atmospheric growth rate shows substantial interannual variability (Ciais et al., 2013). Currently, estimates of variability in anthropogenic carbon emissions, land use change, and oceanic carbon uptake are insufficient to explain the observed variability of the atmospheric $CO_2$ growth rate, and it is assumed that the poorly constrained terrestrial carbon sink must be highly variable (Ciais et al., 2013). Therefore, a detailed knowledge of the interannual variability of each of the atmospheric $CO_2$ sources and sinks is needed to predict the atmospheric $CO_2$ growth rate, and hence future climate. Interannual variability in the ocean carbon flux, including in the North Atlantic, is not well constrained (Schuster et al., 2013), and so there is a need to improve our understanding of its magnitude and the underlying mechanisms. Here, evidence has been presented to show that interannual variability in North Atlantic temperature and preformed alkalinity modify DIC solubility, which sets the interannual variability of the basin's carbon inventory. Furthermore, those variations can be estimated to first order using observations of ocean temperature and salinity that are much better sampled than DIC and alkalinity.

Other work finds that that DIC inventory variability is the leading driver of interannual changes in the basin's surface ocean $p$CO_2 (Thomas et al., 2008; Doney et al., 2009), and that ocean $p$CO_2 variability is the main driver of North Atlantic flux variability (Couldrey et al., 2016). Clearly, detailed observations of the ocean carbon sink are necessary to accurately constrain ocean $CO_2$ fluxes (and hence, understand the changing rate of growth of atmospheric $CO_2$), however this work finds evidence that much of the interannual variability of North Atlantic DIC inventory can be estimated to first order by quantifying saturation effects. The observations necessary to calculate interannual variability in C'$^{sat}$ are temperature, salinity, and pressure, which are all regularly sampled by free-floating profiling floats in the upper 0-2000 m (Roemmich et al., 2009). Furthermore, there are increasing numbers of profiling floats equipped with oxygen sensors, and progress is being made towards the creation of large scale data sets (e.g. Takeshita et al., 2013), which could be useful to further constrain estimates of North Atlantic DIC variability by estimating the C$^{soft}$ contribution. These findings should be useful in informing the construction of regional carbon budgets and in international emissions monitoring efforts.

Recent estimates of air-sea carbon fluxes at the basin scale (e.g. Rödenbeck et al., 2014; Landschützer et al., 2016; Lauvset et al., 2016) employ some form of mapping technique to estimate the distribution of a sparsely observed parameter (like $p$CO_2) using better observed fields (like temperature, salinity, oxygen, etc). The reliability of these methods is inherently unknowable





without more comprehensive observations of the carbon system. This work shows that variability in temperature, salinity, and oxygen concentrations provides most of the information necessary to capture interannual DIC variability. Therefore, our work demonstrates that approaches that leverage fields of temperature, salinity, and oxygen are likely to produce robust estimates of carbon cycle variability even where carbon observations are rare.

*Data availability.* Model output used in this study is available on request. Owing to the large size of the output (several gigabytes), readers are encouraged to get in touch to discuss the best methods of data transfer. Observation datasets described here are available following details in their respective publications

**Appendix A: Implementation of preformed tracers in the model**

$$\frac{d\mathrm{Tr}^{pre}}{dt} = \frac{\mathrm{Tr}^{pre} - \mathrm{Tr}}{\tau} \tag{A1}$$

Further to the configuration of NEMO-MEDUSA-2 described in the main manuscript and elsewhere (Yool et al., 2013a, b; Couldrey et al., 2016), two preformed tracers were added; preformed alkalinity ($A_T^{pre}$) and preformed dissolved inorganic nitrogen ($\mathrm{DIN}^{pre}$). An ideal preformed tracer's concentration ($\mathrm{Tr}^{pre}$) in the mixed layer is equal to the concentration of the 'real' counterpart (Tr). Away from the mixed layer, the preformed tracers are not subjected to any biogeochemical processes, and instead are only varied through physical redistribution. For the purposes of this work, these passive tracers are needed

to understand the biogeochemical processes that happen deeper than the layer of primary productivity. Therefore, the surface ocean (where the preformed tracers' concentrations are set) is defined as the depth above which 95% of the global mean primary productivity occurs. At the end of 600 years of spinup using 20 repeated cycles of the control forcing set, this depth is calculated as 62m. Above this depth, the preformed tracer concentration, $\mathrm{Tr}^{pre}$ is relaxed toward the total tracer concentration, Tr, using Eq. (A1). The relaxation timescale, $\tau$, is set to 1 day, to quickly adjust the $\mathrm{Tr}^{pre}$ towards Tr.

$$A_{Tsurf} \approx S_{surf} m + c \tag{A2}$$

$$A_T^{pre} = S m + c \tag{A3}$$

To initialise $A_T^{pre}$ and $\mathrm{DIN}^{pre}$ in the model, fields of both tracers were estimated empirically using the model state at the end of 600 years of spin-up. If salinity is assumed to be conserved in the ocean interior, it possible to estimate $A_T^{pre}$ by linearly regressing surface alkalinity ($A_{Tsurf}$) against surface salinity ($S_{surf}$), and then using the coefficients $m$ and $c$ with the interior

salinity ($S$), Eq. (A3). The $A_T^{pre}$ field was generated using Eq. (A3) on the model state after 600 years. The $\mathrm{DIN}^{pre}$ initial field was generated by multiplying the model Apparent Oxygen Utilization (AOU) field at the end of the 600 years by a Redfield ratio of nitrogen to oxygen ($R_{NO} = -16/151$, (Yool et al., 2013a)). The model was then run for a further 300 years (10 forcing cycles) with the preformed tracers included. It was not practical to run the model for the 1000s of years needed to reach



physical or biogeochemical equilibria. However, drifts in the upper 2000 m (where this study's variability of interest occurs) were judged to be sufficiently small after these 900 years of spin-up.

**Appendix B: Partitioning components of DIC using preformed tracers**

One may express the DIC concentration of a water parcel, C, as the sum of preformed ($C^{pre}$) and regenerated ($C^{reg}$) carbon, Eq.

(B1). 'Preformed' carbon describes the dissolved inorganic carbon concentration that seawater had when it was last ventilated at the surface, and 'regenerated' carbon refers to the DIC that has been subsequently added into solution. The term 'regeneration' is used here to refer to the process by which any tracer (nutrient, carbon or alkalinity) becomes enriched (added into solution) away from the mixed layer relative to the 'preformed' tracer concentration that is set in the mixed layer. Strictly speaking, the biogeochemical processes that cause nutrients, carbon, and alkalinity to enter solution beneath the mixed layer are all distinct,

but can be considered analogous to nutrient regeneration, since they all have the effect of enriching a tracer concentration above its preformed concentration. If a water mass is in contact with the atmosphere (i.e. it is in the surface mixed layer) then its preformed and total tracer concentrations are the same, and the regenerated component is zero.

$$C = C^{pre} + C^{reg} \tag{B1}$$

Preformed carbon can be partitioned into saturation ($C^{sat}$), disequilibrium ($C^{dis}$) and anthropogenic ($C^{anth}$) components,

Eq. (B2). $C^{sat}$ is calculated with standard carbonate system equations (van Heuven et al., 2011) and is a function of a water parcel's temperature, salinity, the mixing ratio of $CO_2$ in the preindustrial atmosphere ($X^{pre}_{CO_2}$=286 ppmv), and the alkalinity it had when last in contact with the surface (its preformed alkalinity, $A^{pre}_T$), Eq. (B3). $C^{dis}$ refers to the excess or deficit of carbon that a water parcel may have if it is subducted away from the mixed layer without being saturated relative to the preindustrial atmosphere. The $C^{dis}$ reservoir is distinct from the excess carbon content of seawater due to anthropogenic activities, $C^{anth}$.

The calculation of $C^{dis}$ and $C^{anth}$ are described at the end of this section.

$$C^{pre} = C^{sat} + C^{dis} + C^{anth} \tag{B2}$$

$$C^{sat} = f(T, S, A^{pre}_T, X^{pre}_{CO_2}) \tag{B3}$$

$$A_T = A^{pre}_T + A^{reg}_T \tag{B4}$$

As with carbon, alkalinity ($A_T$) can also be expressed in terms of preformed ($A^{pre}_T$) and regenerated ($A^{reg}_T$) components, Eq.

(B4). Preformed alkalinity is set up as a passive tracer in the model, as described in Appendix A. This passive tracer is required



in ocean simulations experiencing transient climate change, since empirical approaches to estimating $A_T^{pre}$ break down if the climate is not at steady state (Williams and Follows, 2011; Bernardello et al., 2014).

$$C^{reg} = C^{soft} + C^{carb} \tag{B5}$$

$$C^{soft} = -R_{CN}(DIN - DIN^{pre}) \tag{B6}$$

$C^{reg}$ in Eq. (B1) represents the contribution of the soft tissue pump ($C^{soft}$) and the carbonate pump ($C^{carb}$) to DIC, Eq. (B5). Soft tissue carbon ($C^{soft}$) therefore describes the DIC that is added to seawater by organic matter respiration at depth. To diagnose $C^{soft}$ a second preformed tracer is used: preformed DIN ($DIN^{pre}$). The biogeochemical model, MEDUSA-2, considers DIN as an aggregation of all inorganic nitrogen species, and does not separately represent nitrate, ammonium, nitrite etc. (Yool et al., 2013a). This simplification means that biogeochemical calculations are performed using a bulk DIN pool,

rather than accounting for differing behaviour of nitrogen species. If one assumes that carbon is regenerated at depth according to a fixed stoichiometry with nitrogen, then preformed DIN may be used to estimate $C^{soft}$ using Eq. (B6). $R_{CN}$ is the Redfield ratio of carbon to DIN, 106:16 in the model (Yool et al., 2013a), after (Anderson, 1995), and DIN and $DIN^{pre}$ are total and preformed DIN concentrations respectively. In this model setup, $C^{soft}$ is therefore created away from the ocean surface (because at depths shallower than 66 m, $DIN^{pre}$ = DIN and $DIN^{reg}$ = 0). When any $C^{soft}$ is vertically transported into the

upper 66 m, it loses its identity as $C^{soft}$ and is converted into $C^{dis}$.

The carbonate pump contribution ($C^{carb}$) represents the DIC that has been added to seawater as a result of the dissolution of $CaCO_3$. For every mole of $CaCO_3$ dissolved, one mole of carbon is regenerated (added to the DIC pool as $C^{carb}$) alongside one mole of calcium ions, $[Ca^{2+}]$, Eq. (B7). The addition of DIC to seawater in this way can be quantified from changes in seawater alkalinity away from preformed values. Since calcium ions are one of the components of alkalinity and are doubly

charged, for each mole of calcium ions dissolved, there is a change in $A_T$ ($\delta A_T^{CaCO3}$) by two moles, Eq. (B8).

$$C^{carb} = \delta[Ca^{2+}] \tag{B7}$$

$$\delta A_T^{CaCO3} = 2\delta[Ca^{2+}] = 2C^{carb} \tag{B8}$$

Organic matter respiration also regenerates alkalinity at depth through the release of nitrate ions, $[NO_3^-]$, into seawater and must be accounted for, Eq. (B9). The effect of the production of nitrate ions by respiration on alkalinity ($\delta A_T^{NO3}$) can be

calculated by multiplying a fixed Redfield ratio of nitrate released to oxygen consumed ($R_{NO}$ = -16/151) by the AOU, Eq. (B10).

$$\delta A_T^{NO3} = -\delta[NO_3^-] \tag{B9}$$

$$\delta A_T^{NO3} = R_{NO}AOU \tag{B10}$$



The total change of $A_T$ at depth is therefore the regenerated alkalinity and is the sum of the effects of both $CaCO_3$ dissolution and organic matter respiration, Eq. (B11). By substituting Eq. (B8) and (B10) into Eq. (B11), one arrives at an equation for $A_T^{reg}$ in terms of $C^{carb}$ and AOU. According to the decomposition of alkalinity in Eq. (B4), $A_T^{reg}$ is the difference between seawater $A_T$ and $A_T^{pre}$, Eq. (B13). By substituting (B13) into (B12) and rearranging, an equation for $C^{carb}$ in terms of regenerated alkalinity and nitrogen is yielded, Eq. (B14). However, in MEDUSA-2's simplified representation of alkalinity cycling, nitrate remineralisation does not affect $A_T$ (the computational cost of this effect was considered to outweigh the gain in accuracy) and so this term does not feature in the final equation for $C^{carb}$, Eq. (B15). As with $C^{soft}$, when $C^{carb}$ is upwelled into the upper 66 m, it loses its identity as $C^{carb}$ and becomes $C^{dis}$.

$$A_T^{reg} = \delta A_T^{\mathrm{CaCO3}} - \delta A_T^{\mathrm{NO3}} \tag{B11}$$

$$A_T^{reg} = 2C^{carb} + R_{\mathrm{NO}}\mathrm{AOU} \tag{B12}$$

$$A_T^{reg} = A_T - A_T^{pre} \tag{B13}$$

$$C^{carb} = \frac{1}{2}(A_T - A_T^{pre} - R_{\mathrm{NO}}\mathrm{AOU}) \tag{B14}$$

$$C^{carb} = \frac{1}{2}(A_T - A_T^{pre}) \tag{B15}$$

An estimate of the excess oceanic DIC due to anthropogenic carbon additions ($C^{anth}$) is found by subtracting the DIC field of the Anthropogenic run at a given time, $t$, ($C_t^{AN}$) from that of Warming Only run ($C_t^{WO}$), Eq. (B16). This is because the Warming Only run adjusts to climate change physically (i.e. it undergoes global warming) but does not accumulate additional oceanic carbon. The difference between the Anthropogenic and Warming Only runs is therefore the excess carbon added to the oceans due to rising atmospheric $p$CO$_2$.

$$C_t^{anth} = C_t^{AN} - C_t^{WO} \tag{B16}$$

By substituting Eq. (B2) and (B5) into Eq. (B1), a complete decomposition of DIC can be written, Eq. (B17). By subtracting the derived fields of $C^{sat}$, $C^{soft}$, $C^{carb}$ and $C^{anth}$ from modelled fields of C, an equation for disequilibrium carbon emerges, Eq. (B18). In similar decompositions of DIC (Williams and Follows, 2011), preformed alkalinity must be estimated empirically, AOU is used to estimate some of the components, and $C^{anth}$ cannot be easily quantified. The result is that when using an observation-based empirical DIC decomposition, it is not possible to distinguish $C^{anth}$ and $C^{dis}$, nor the errors that accumulate using such approaches. By using explicitly simulated preformed tracers in numerical ocean models, it is possible to determine all the right hand side terms of Eq. (B18) and solve for $C^{dis}$, and the major errors that arise from assumptions about empirical relationships are avoided.

$$C = C^{sat} + C^{soft} + C^{carb} + C^{dis} + C^{anth} \tag{B17}$$

$$C^{dis} = C - (C^{sat} + C^{soft} + C^{carb} + C^{anth}) \tag{B18}$$





**Appendix C: Root-Mean-Square Calculation**

$$RMS(\mathrm{X}) = \sqrt{\overline{(\mathrm{X} - \bar{\mathrm{X}})^2}} \tag{C1}$$

The root-mean-square (RMS) is a useful measure of the magnitude of variance of a time series. For any given water property, X, the long-term time mean ($\bar{\mathrm{X}}$) can be calculated, then the difference of each annual mean from the long term mean may be found ($\mathrm{X} - \bar{\mathrm{X}}$). These anomalies are then squared, and the mean squared anomaly calculated. Finally, the square root of the mean squared anomalies is then calculated to yield the temporal RMS, Eq. (C1).

**Appendix D: Empirical estimation of components of DIC**

Section 4.4 in the main manuscript describes results employing a commonly used empirical decomposition of DIC (Volk and Hoffert, 1985; Ito and Follows, 2005; Williams and Follows, 2011), rather than the modelled tracer-based method used everywhere else in this work. The theory behind the empirical method is described in Appendix B, and this section provides details of its implementation to generate the results in Section 4.4.

In order to set up an initial field of the preformed tracer, $A_T^{pre}$, in the model, an empirical estimation technique using the linear relationship between total alkalinity and salinity at the surface was described in Appendix A. This same method is also used to estimate modelled variability in $C^{sat}$. The relationship used to estimate interior preformed alkalinity from salinity was calculated using annual mean modelled surface fields spanning 1980-2019, with coefficients shown in Eq. (D1), an $r^2$ value of 0.73, Eq. (D2) an RMSE of 26.0 mmol m$^{-3}$, Eq. (D3).

$$\begin{aligned}
\mathrm{A}_T^{pre} &= S \times 55.64 + 412.88 \tag{D1}\\
r^2 &= 0.73 \tag{D2}\\
RMSE &= 26.0 \tag{D3}
\end{aligned}$$

The estimates of modelled $C^{soft}$ variability presented in Section 4.4 were calculated using Apparent Oxygen Utilization (AOU), and not the explicitly modelled tracer of DIN$^{pre}$. To do this, the same theoretical approach was taken that was used also to initialise the model field of preformed nutrient (DIN$^{pre}$). To estimate $C^{soft}$ without an explicit model tracer of preformed nutrient, the model fields of AOU are multiplied by R$_{\mathrm{CO}}$, the remineralisation ratio of carbon to oxygen, Eq. (D4), where R$_{\mathrm{CO}} = -106/151$ (Yool et al., 2013a). Thus, for every mole of oxygen consumed through respiration, R$_{\mathrm{CO}}$ moles of DIC are added to seawater.

$$C^{soft} = \mathrm{R}_{\mathrm{CO}} AOU \tag{D4}$$

Similarly, the estimates of modelled $C^{carb}$ variability presented in Section 4.4 were calculated using the linear regression of preformed alkalinity onto salinity, and not the explicitly modelled tracer of $A_T^{pre}$. $C^{carb}$ is then estimated using Eq. (B15).





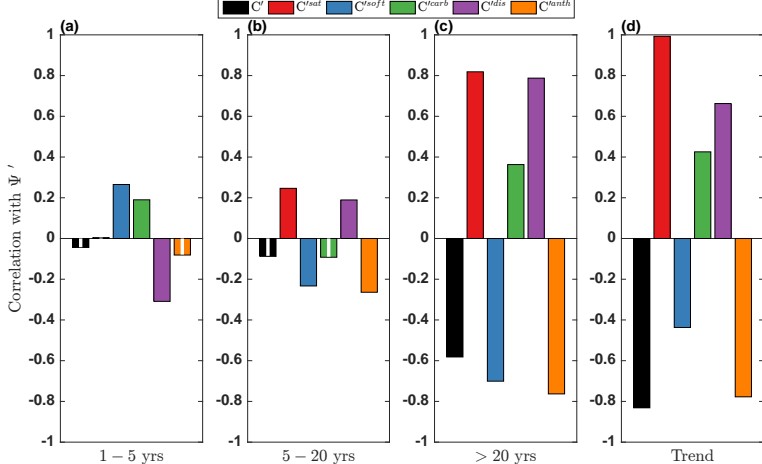

**Figure F1.** Correlations of Atlantic Meridional Overturning strength with DIC components filtered for different timescales of variability: interannual (1–5 years, a), decadal (5–20 years, b) and multidecadal (>20 years, c), and the long term trend (d). Bars with a white line through the middle denote correlations insignificant at 95% confidence (i.e. with p-values greater than 0.05)

## Appendix E: Long term carbon inventory changes due to climate change

$$I_t = \int\limits^{basin} C_t \tag{E1}$$

$$\Delta I = \overline{I_{2090:2099}} - \overline{I_{1860:1869}} \tag{E2}$$

$$\Delta I^{CC} = \Delta I^{AN} - \Delta I^{CN} \tag{E3}$$

5    The bulk basin changes in each carbon component due to climate change, are calculated as follows: starting with the annual mean basin inventory of each carbon component ($I$) for any given year ($t$), Eq. (E1), the difference between the mean inventories of the first and last decades (1860 to 1869 and 2090 to 2099 respectively) is found ($\Delta I$), Eq. (E2). The inventory change due to climate change ($\Delta I^{CC}$) is calculated as the difference between $\Delta I$ for the Anthropogenic (AN) simulation and the Control (CN) simulation, Eq. (E3), to account for any drifts in the model.

## 10    Appendix F: Correlation of DIC variability with various North Atlantic ocean properties

In this section, the time series of basin total DIC component variability is correlated with different phenomena to shed light on which large scale oceanographic processes are likely to drive the modelled interannual carbon cycle variability. Since strong correlations between obvious candidate processes were not found, this analysis mainly serves to discount potential mechanisms.

The Atlantic Meridional Overturning Circulation (AMOC) strength was found by first calculating the along-J-coordinate

15   (i.e. effectively meridional) overturning streamfunction at 26.5° N, and then taking the maximum streamfunction value below



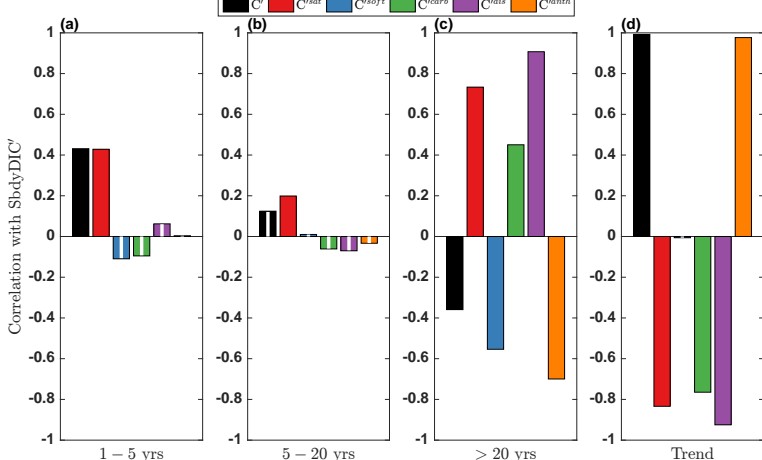

**Figure F2.** Correlations of mean DIC concentration at the open southern boundary of the North Atlantic domain (the equator) with DIC components filtered for different timescales of variability: interannual (1–5 years, a), decadal (5–20 years, b) and multidecadal (>20 years, c), and the long term trend (d). Bars with a white line through the middle denote correlations insignificant at 95% confidence (i.e. with p-values greater than 0.05)

a depth of 500m. The correlation analysis was also performed using the maximum streamfunction value below 500m from anywhere in the basin (rather than exactly at 26.5° N) but the results were qualitatively the same and are not included here. Clearly the AMOC is strongly linked with multidecadal and the long term Atlantic DIC variability for all components (Fig. F1c,d), but no strong correlations were found for the interannual or decadal variability (Fig. F1a,b).

The whole-basin DIC component inventories were also correlated against a time series of the mean DIC at the domain's southern boundary, the equator (Fig. F2). The interannual C and $C^{sat}$ variabilities are moderately correlated with the DIC at the southern boundary. Since much of the interannual C variability is driven largely by $C^{sat}$, the correlations for these two components are similar, indicating that the flux of saturation carbon across the equator plays a role in the total basin DIC variance, but is not the main driver of variability at the basin scale.

Finally, the DIC component inventories were correlated against the basin mean temperature (Fig. F3). The interannual $C^{sat}$ is strongly (r = -0.75) anticorrelated with the basin mean temperature, and so clearly the basin's heat content is a key factor controlling saturation variability (Fig. F3a). However, average basin temperature is weakly anticorrelated with interannual DIC variability (Fig. F3a).

## Appendix G: Trend removal and Fourier analysis

Fourier analysis on finite length signals using the discrete Fourier transform assumes that the signal is perfectly periodic (i.e. the energy at the start and ends of the signal are identical). If the spectrum of a time series is calculated from a time series





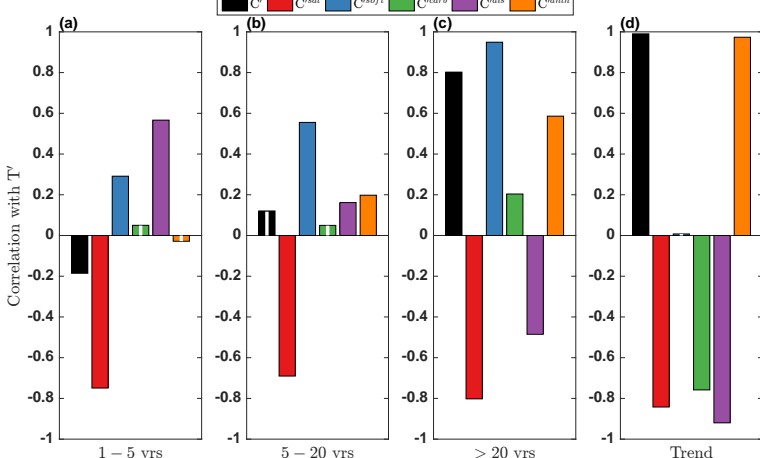

**Figure F3.** Correlations of North Atlantic basin mean temperature with DIC components filtered for different timescales of variability: interannual (1–5 years, a), decadal (5–20 years, b) and multidecadal (>20 years, c), and the long term trend (d). Bars with a white line through the middle denote correlations insignificant at 95% confidence (i.e. with p-values greater than 0.05)

that contains a trend, then the Fourier transform will reflect this by including a spurious amplitude at every period/frequency. These false amplitudes across all periods stem from the violation of the assumption that the finite length signal is periodic. Consider the spectra of two identical time series, which have identical variability on all time scales, except one has a trend and the other does not. If one were to calculate and plot the spectra of both without removing trends, then the spectrum of

the time series containing a trend would look identical to the other, except it would be vertically displaced upwards towards higher amplitudes across all periods. The vertical displacement of the spectra then makes it difficult to compare which periods of variability are common to both spectra. If the trend of a time series is not well described by a linear function, then some residual trend may remain in the time series after detrending. Taking the time derivative of a time series helps to detrend the signal, while still preserving variability of interest, and residual trend will be smaller. This makes the assumption that the finite

signal is perfectly periodic closer to being valid, and the calculated spectrum more meaningful. In practice, when the effects of residual trends are smaller than the variability of interest, then the spectra become aligned closely enough that they can be compared (like the spectra in Fig. 2). Note that the assumption of periodicity is not required for the rest of the statistical variability analysis described in this manuscript, and so trend removal using polynomial functions is a valid approach. The strengths and weaknesses of both approaches to variability analysis are complementary.

*Author contributions.*   TEXT



*Competing interests.* The authors declare that they have no conflict of interest.

*Disclaimer.* TEXT

*Acknowledgements.* This work was supported the RAGNARoCC NERC directed research programme (NE/K002546/1, NE/K00249X/1 and NE/K002473/1). The observational data used are available via the relevant references that accompany them in the text. The authors are

5   very grateful to those individuals and institutions who have produced and made freely available the GLODAPv2 dataset used in this work; (doi:10.5194/essd-8-297-2016). The authors are particularly grateful to Dorothee Bakker and Bablu Sinha, who have provided helpful and constructive feedback on this work.



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
