# Peer review of "Drivers of 21st Century carbon cycle variability in the North Atlantic Ocean"

_Biogeosciences, 2019_

## Referee Comment (RC1) · Anonymous Referee #1 · 16 Feb 2019

A. Summary and recommendations

In this manuscript, the authors assess the timescales of carbon cycle variability in the North Atlantic region of a numerical ocean circulation model under global warming conditions. By partitioning the carbon inventory into constituents associated with different physical and biogeochemical processes, the authors attempt to discern the drivers of variability on timescales from inter-annual to multi-decadal. The main result is that inter-annual variability in the carbon inventory is driven by changes in the saturation state of carbon, which itself arises from changes in temperature and preformed alkalinity. Other processes, including the biological carbon pump, play a more significant role on decadal to multi-decadal timescales, and the long-term trend is established by the uptake of anthropogenic carbon.

[Figure]

The manuscript is well-written, and the figures well-presented. Overall, the analysis is interesting, but in my opinion incomplete. In particular, the authors have not taken sufficient advantage of the benefits conferred in using a numerical simulation, whereby the full carbon budget can be determined. Instead (unless I am mistaken) they evaluate variability only in the storage terms of total inorganic carbon and its constituents, leaving the author to wonder whether this reflects drivers of air-sea exchange or fluxes across the open domain boundaries – which appears to me to be of critical importance in interpreting the results. I elaborate on this issue below, and invite the authors to correct me if my understanding is mistaken. This concern, in combination with other issues outlined below, means that although the work inspires some intrigue, it has too many 'open doors' to provide a robust increase in our understanding of NA carbon cycle variability and thus to be considered ready for publication at this stage. I would invite the authors to consider resubmission after a more comprehensive budget analysis has been carried out.

B. Major issues and considerations

B.1. Linking inventory changes with variability in air-sea exchange.

As noted in the summary, I have a substantial concern with regards to the absence of a closed budget analysis. If the goal is to assess the drivers of ocean carbon uptake variability (which I perceive as the main thrust of the study), variations in the basin inventory of total carbon and its constituents confers relevant information only when the variability in lateral fluxes of carbon across the domain boundaries are either evaluated or considered negligible. As far as I can tell from the manuscript, neither of these has been done or shown. The authors comment on page 2, lines 9 to 11 (p2:9-11) that "variability in oceanic carbon concentration is a leading-order control on the amount of $CO_2$ that the oceans absorb", and offer references in support of this. However, this offers little information about the timescales or regions over which this is case.

It is plausible that changes in the carbon transport across lateral boundaries plays

a non-negligible role in the variability of the basin-wide carbon inventory. A back-of-the-envelope calculation suggests a magnitude for the net carbon transport at either the northern or southern boundary on the order of 0.01 PgCyr-1 (volume transport of 10 Sv, mean seawater density of 1000 kgm-3, mean total inorganic carbon difference between inflowing and outflowing waters of 20 $\mu$molkg-1). I suspect that this is a rather conservative estimate, and I would invite the authors to calculate the actual mean flux in the model. It is further feasible that, through volume transport changes alone, this could vary by as much as 100% on inter-annual timescales and longer (e.g. inter-annual variability at 26N as observed by the RAPID array; I am unsure if the same extent of variability also exists in the model, or at the equator where is the southern boundary of the analysis domain). Figures 2a and 3 indicate that changes in the carbon inventory are in the order 0.1 PgCyr-1 and less (except for the long-term trend). In my interpretation, these numbers indicate that convergence/divergence of carbon in the basin by transport across the lateral boundaries at least cannot be excluded a priori as a source of basin-wide carbon inventory change.

This convergence/divergence can of course be readily calculated in the model, and thereby easily ruled out as a source of variability. Or, if it is not negligible, it can be subtracted from the inventory to leave air-sea carbon exchange as an inferred residual (indeed the air-sea exchange could also be evaluated in the model, to ensure that the carbon budget, and that of the constituents, are in balance). It strikes me that this is an appropriate and necessary step to take before valuable information can be derived from the analysis. I am confident that the authors are aware of this, and am thus left unsure as to why they appear to have abstained from incorporating a full carbon budget closure as a key part of their analysis. This leaves me to wonder whether there is something that I have missed, or misunderstood. For example, perhaps a closed-budget analysis of the carbon constituents is not feasible because of using differences between simulations to get some of the terms and/or what is done to remove the model drift (though I cannot see clearly why this would be the case). If I am mistaken in my assessment, I would invite the authors to clarify their reasoning and their approach,
pointing towards the analysis or literature that justifies neglecting variability in lateral transports. Without such a justification, the results are left unsatisfactorily open and challenging to interpret. For example, it may well make sense that inter-annual changes in surface temperature drive changes in carbon saturation resulting in carbon uptake variability, but the absence of a closed budget prevents one from directly making this assertion.

B.2. Redundant analyses.

Section 4.4, which explores the correlation between saturated and total carbon appears to me to be somewhat of a repetition of parts of Section 4.2, in which total carbon is shown to be strongly correlated to saturated carbon on inter-annual timescales. The only difference that I can perceive is that, in the latter case, preformed alkalinity is inferred from its relationship to salinity. Thus, under the guise of showing the explanatory power of saturated carbon variability (Figure 7b), you are simply repeating the assertion that these variables are correlated (Figure 5b) and additionally that an approximated preformed alkalinity has variability consistent with the 'actual' preformed alkalinity as determined in the model. As noted in the text (p15:30-31), this latter point is of some value with regards to the observation of carbon inventory variability in the ocean, but in my opinion (unless I am missing something critical about the information conferred in Section?) does not merit the protracted analysis of Section 4.4 – that is to say, there would be easier and more robust ways to show this. Further showing the modest change in explanatory power afforded by the inclusion of soft tissue carbon and dissolved carbonates is again a repetition of the information conferred in Figure 5b.

I wonder if a more valuable analysis would be to fold aspects of Section 4.4 into the previous section, by considering how variations in temperature and preformed alkalinity change the explanatory power of saturated carbon? As explained further below, I am still struggling to fully understand how the two components interact.

B.3. Spatial distribution of variability.

I found Figure 4, showing the spatial distribution of carbon inventory variability, to be rather insightful. Although caution in its interpretation is advised by the authors (p11:7-9), it still strikes me that a lot can be inferred about the potential driving processes of carbon uptake from these maps. [As an aside, I could not find anywhere an explanation for how these maps were constructed in terms of separating out the timescales of variability. I presume that the column inventories were band-pass filtered.] While the authors list hypotheses (p11, paragraph beginning line 11), they do not describe what aspects of Figure 4 lend themselves to these interpretations. Indeed, this list of hypotheses could have been drawn up even before the simulations were run. It would be valuable for the authors to elaborate more fully on what the spatial distributions tell us about the most likely sources of variability on different timescales. In line with this, it seems like an obvious further step to reproduce Figure 4 for the different constituents of the carbon partitioning. Even though one cannot definitively attribute spatial changes to basin-wide variability, there is much more to be learned from these distributions about the processes establishing carbon uptake variability.

Have the authors considered producing maps such as this in density-latitude space (that is, taking the zonal integral of carbon and constituents in density bands)? I suspect that this would confer substantial further information about the driving processes of variability on different timescales. For example, the density (or depth) range in which variability in soft tissue carbon (combined with carbon disequilibria) emerges, may reveal whether these changes reflect changes in production or export (p12:1-2), or indeed subsurface remineralization.

B.4. Robustness and clarity of statistical analysis.

The statistical analysis in the paper is straightforward, and thus commendably easy to follow. However, it hard to countenance in the absence of any tests of robustness, or reference to literature on time-series analysis. This is not a subject area that I know

well, but I presume that there is a rich literature on the attribution of variability to co-varying components, including separation in time bands. Is the assessment of linear correlations and root-mean-square deviations an appropriate and robust method for doing this? If so, are there caveats or intricacies in the interpretation? Note that I am not criticizing the authors' approach off-hand, but simply highlighting that in my opinion the reader is given insufficient information or detail to convince themselves of the scope and implications of the analysis. An example of where uncertainty arose for me, is in the attribution of multi-decadal variability (Figure 5g, h, i). The authors note that saturated, soft tissue and anthropogenic carbon all have an RMS comparable to total carbon, high correlation (with saturated carbon negatively correlated), and strong linear gradients. The authors conclude that "large variability in Csat and Csoft components is approximately compensatory, and variability in the accumulation of anthropogenic carbon dominates the variability of the North Atlantic carbon sink" (p14:6-8). Is there an aspect of the statistical analysis that precludes an alternative interpretation, that variability in the saturation state compensates anthropogenic carbon uptake, and that biological drawdown establishes multidecadal variability in the carbon sink. I have the impression that either hypothesis is difficult to assert with the present analysis, but invite the authors to provide more detail on why that might not be the case.

B.5. Model validation.

I'm afraid that the authors' model evaluation did not convince me that "the model can be judged to be reasonable and our setup is fit for this study". I should state up-front that I am not a pedant for perfect model-observation matching. Simulations can of course look distinct from observations but still provide valuable insight. However, if the authors wish to infer (as it appears from the writing of the manuscript) that their results inform our understanding of ocean processes (rather than simply those operating in the model [understanding of which can valuable on its own]), specific components of the model-observation comparison should be robust. The authors have made some attempt to explore this but, in my opinion, have so far failed to sufficiently show that the simulations

are fit for purpose. Most crucially, the thrust of the authors' approach is to assess the temporal variability of carbon inventories. It does not make sense to me therefore, why in their statistical comparison with GLODAP data (Figure 1), they include both spatial and temporal variations in the frequency distribution (panels a and d). Perhaps there is something that I have missed, but I cannot see that this provides any relevant information on how reliably the model reproduces ocean carbon variability. Indeed, I suspect that most of the distribution in Figures 1a and d reflects spatial variability, with some component associated with seasonality in the upper ocean. [As an aside, the x-range of the axis in panels a and d should be narrowed to show clearly the appropriate range.] This leaves me unsure what such a distribution actual informs us about the utility of the simulation. Indeed, it could be the case that the mismatch between models and observation in this frame is almost entirely due to spatial bias, and that the models captures the temporal variability well, but I cannot infer this from the present analysis. Of course, the GLODAP data is limited in what information it can provide about temporal variability (although decadal trends, and repeat sections, may be of some value). Comparison with data such as the HOT or BATS time-series, although not necessarily straightforward as the authors note (p6:28-30), would provide more robust assessment of how the timescales of carbon cycle variability in the model match those of the real world.

B.6. Inter-annual variability of preformed alkalinity.

In Section 4.3, the authors show that inter-annual variability in saturated carbon arises from changes in both temperature and preformed alkalinity. It is clear to see what processes might drive temperate variations. What I am left wondering is what process establishes inter-annual variability in preformed alkalinity? Given the apparently small variability in biology on these timescales, I can't believe that this arises from changes in the formation/dissolution of carbonates or the formation/respiration of organic matter. Is this then a signature of variations in dilution/concentration, presumably by inter-annual precipitation/evaporation? Or is this the result of convergence/divergence of preformed

alkalinity at the lateral boundaries? On the other hand, is it simply a quasi-spurious result of the statistical analysis that requires more careful interpretation? These questions are left inadequately explored in the study, and I was left unsure how to interpret the analysis.

C. Conclusions

In conclusion, while this study provides some potentially interesting results, the analysis remains incomplete. In particular, the approach leaves too many open-ended questions for reliable conclusions to be drawn about the operation of the North Atlantic carbon sink, either in the model or in the ocean. Some substantial but relatively straightforward steps (such as the closure of the basin-wide carbon budget) could be taken to improve the reliability and ease of interpretation of results. I would advise the authors to explore a more robust, rounded approach, and then consider resubmission.

---

## Referee Comment (RC2) · Anonymous Referee #2 · 5 Apr 2019

In this study the ocean carbon pool is partitioned into different contributions (anthropogenic, soft, carbonate, saturated and disequilibria) and the significance of these contributions on controlling variability in different timescales in the North Atlantic is investigated using a realistic numerical model. The highlighted conclusion is that on interannual timescales the variability is driven by the effect of the ocean temperature and preformed alkalinity changes to the ocean uptake of carbon, while on decadal-multidecadal timescales the changes in the soft tissue pump also become important. The anthropogenic carbon component is not driving the variability on interannual timescales but becomes increasingly important on following timescales and dominates on long timescales.

The study is interesting, worthy of publication and fits the scope of the journal. The

results are well supported by statistical analysis, strengthening the authors arguments. The statistical methods are described in detail and the overall manuscript reads well. However, to me, in places there is too much focus on the statistics and repetition, rather than a focus on the insight gained from the analysis. I also believe that the figures showing the correlations between different processes and the different carbon contributions should be presented in the main text rather than be buried in Appendix F. Further discussion on the different mechanisms that lead to these correlations is also necessary in my opinion, as I personally find this part of the study to be very interesting. Finally, although the manuscript reads well, I believe that a re-organisation of Section 4 would benefit the manuscript. Based on the above and some other issues discussed in my following comments, I recommend the following revisions/concerns be addressed before the manuscript be accepted for publication.

Major comments:

1. Section 4 is to me somewhat repetitive and in need of re-organisation. I suggest that the whole section is re-organised into discussion driven by the timescales such that there are 3 subsections (interannual variability, decadal-multidecadal variability, and long-term variability).

1.a. Section 4.1: I found some of the statistical analysis repetitive and unnecessary. In my opinion, the timescales decomposition using the FFT, and so Figure 2b, do not add any new insight that cannot be derived by the following analysis using equations 2-6. Therefore, I think this analysis is redundant and that the associated text, Appendix G and Figure 2b should be removed. Instead, I suggest that the authors expand somewhat on the discussion concerning the analysis using equations 2-6 so that all the carbon contributions are shown for the different timescales along with the total in each panel of Figure 3. Else if the authors do not want to crowd the figure, they may repeat this figure for the different carbon contributions. I believe this visual comparison is intuitive and will complement the results from statistics in Figure 5.

[Figure]

1.b. Likewise, in my opinion, Figure 4 should be repeated for all the constituents. As it is now, I gain no insight from the text on page 11 last paragraph continue to page 12. In this paragraph the authors simply suggest plausible processes/hypotheses. Instead, I believe that the authors should isolate which of these hypotheses may apply to their results based on the maps of Figure 4 repeated for all the carbon constituents. Then the statistical analysis will complement, refine and confirm the insights gained by these maps.

1.c. Section 4.4 "Role of Csat in interannual DIC variability", to me this discussion is somewhat redundant. In my opinion the authors already explained that Csat, and so solubility and preformed alkalinity dominate the interannual variability. As I understand, the novelty of this section is the empirical estimates of preformed alkalinity from salinity, and Csoft from AOU to demonstrate that temperature and salinity observations can to first order be used to evaluate the DIC variability, with the AOU observations having the potential to further improve this evaluation. Hence, I suggest that the authors substantially shorten this discussion. If the authors follow my advice about reorganising section 4 into 3 subsections driven by timescales, this section will naturally merge with section 4.3 and the previous discussion about the interannual timescale into one section.

2. Section 5 and Appendix F. In my opinion, the figures in Appendix F are interesting and I think that they can be highlighted better if they are in the main text (merge F1-F3 as a single figure in section 5). Subsequently, I suggest that these correlations and the mechanisms/processes that drive them are discussed further in Section 5. Although some of the correlations are explicitly explained (AMOC-Csoft), others are only discussed implicitly or not at all. For example, why is the correlation between AMOC and Cdis positive only on decadal and longer timescales (Figure F1)? Why is the correlation between open boundary and Cdis positive on multidecadal timescales but negative on longer time scales (Figure F2)? I suggest that at least the sign of the significant correlations in Figures F1-F3 is discussed and explained.

Minor comments:

3. In the Abstract, line 10, "A mixture of saturation and anthropogenic drivers": I suggest that the authors rephrase since the anthropogenic drivers may be misinterpreted as both due to changes in atmospheric CO2 and in climate (e.g., temperature); but the later is accounted in the saturated part in this study. Maybe instead use "... of saturation and carbon uptake driven by the anthropogenic increase in atmospheric CO2", or something equivalent.

4. Page 4, starting paragraph in line 7: "Other work has investigated in detail ...other work has focus on the spatially varying ...". Which other work? Does this refer to the studies referenced in the previous paragraph? Maybe the authors can clarify or reference here this other work.

5. Page 4, lines 17-19: "The main hypothesis is that oceanic uptake...". I am a little confused by this statement. Is this hypothesis established by previous studies? Is it an a priori assertion of the authors? If the later, I find it a little confusing that this hypothesis does not match the study's results. In my understanding, contrary to what is stated in this hypothesis, the study shows that interannual variability is driven mainly by the saturated component rather than the saturated component and soft tissue pump, and decadal variability is dominated by the saturated part rather than the anthropogenic part. Maybe the authors should rephrase or use a more generic hypothesis like: The main hypothesis is that different processes will drive variability on different timescales.

6. Page 5, line 28 "to distinguish physical from the biogeochemical ocean carbon cycle..." and subsequent use of "physically adjust" in line 31. In my opinion the use of physical here is not accurate. To me the physical part would be the part driven solely by the changes in physical processes due to warming e.g. circulation, stratification. The terms "physical" and "physically adjust" as they are used include the effect from the solubility changes due to changes in temperature and the effect of changes in alkalinity which I would consider as a chemical rather than a physical process. I would suggest to please rephrase.

[Figure]

7. Page 5, line 33: "non-steady state anthropogenic carbon ..". I am a little confused here, I think that in your warming only run the carbon inventory changes would rather be associated with the non-steady state of both anthropogenic and natural part as described in McNeil and Matear (2013). I may be mistaken but please can you clarify.

8. Page 6, subsection 2.2. Are the Csoft and Ccarb the same in both the warming only and the anthropogenic simulations? If not, then to me it seems that some of the changes in biology would be accounted for in the Canth. Is this correct? Can you please clarify?

9. Page 7, Figure 1. Please can you add in the figure's caption that the right panels are for the GLODAP climatology and an equivalent "climatology" from the model rather than the unmapped database.

10. Page 11, Figure 3. Is this figure showing the root-mean square of DIC anomalies of long-term mean as stated in the caption? I think this is a typo and this figure shows the filtered carbon inventory time series following equation 2-6. If not, then I am confused as to how this root mean square is estimated and what it represents? Please can you clarify?

11. Page 20, line 25 ". . . can be estimated to first order by quantifying saturation effect". I understand that the authors mean their saturation component here but since this is the conclusion section, I think that they should be more explicit. Saturation effect in general in my opinion includes both changes due to atmospheric CO2, temperature, and alkalinity which may cause confusion. Maybe rewrite to something more like " . . .to first order by quantifying the effect of solubility and alkalinity"

12 Typos: page 8, line 8: alkalinity is misspelled; page 21, line 23: an "is" is missing such that " . . . interior, it is possible to estimate Apre. . ."

---

## Author Comment (AC1) · 3 May 2019

Response to Reviewer 1:

A. Summary We are grateful for the thorough and engaged review that Reviewer 1 has provided. We agree that the manuscript would benefit from some further analyses, especially regarding a closed budget approach. In addition, we aim to explore our results in more detail, as suggested.

B1 We agree that the exchange of DIC and its components across the lateral boundaries of the North Atlantic is worth exploring quantitatively. We have begun this analysis, and agree this it should be incorporated into the work.

B2 While we found it interesting to explore the extent to which it was possible to derive

interannual DIC variability using a very limited set of predictors, Reviewer one is correct that the underlying message of Section 4.4 is the same as Section 4.2. Reviewer 2 also felt that this analysis was repetitive of the point that saturation effects dominate interannual variability. As a result, the section will be removed and relevant parts will be included elsewhere.

B3 We agree that Figure 4 and the text describing the list of hypotheses should be drawn together more closely. The aim of the section was to get an overview of which processes might drive variability at different timescales, using inferences drawn from the hotspots highlighted by the figure. This section will therefore be clarified to more clearly illustrate how particular features of Figure 4 implicate particular hypotheses about the basin as a whole.

B4 We thank the reviewer for asking for this clarification about the inferences drawn from our statistical analysis. Reviewer 1 makes the point that this statistical approach alone does not make it clear how any components of DIC compensate each other when more than two components dominate variability over a particular timescale. The reviewer is correct, and that more information is required to determine, for example, on multidecadal timescales whether there is compensation between just Csat and Csoft or between Csat and Canth (or some mixture of the two). Our approach provides a robust method of attribution of total DIC variability to individual components (since the total variability is exactly equal to the sum of the parts). As a result, most of the assertions of the manuscript are supported by this statistical analysis (for example when one or two components dominate variability as is the case on interannual and decadal timescales, and for the long term trend). As the reviewer correctly points out, it is not possible (with the present analysis alone) to determine which components compensate other components when 3 or more vary substantially. Instead, such statements will be revised to communicate that multidecadal Csoft and Canth variability act in opposition to that of Csat, and that all components are important in setting the total variability.

B5 Comparing observations and model output at the basin scale is challenging because there is no consensus on how best to make like-for-like comparisons when scaling up point observations to the model gridpoint scale, or the other way around. The analysis shown in Figure 1 is useful because it allows for a relatively 'raw' comparison regardless of the spatiotemporal distribution of observations. Quantifying carbon cycle variability at the basin scale from observations like GLODAP is a serious undertaking in its own right, even before any comparison with model output is attempted. Nevertheless, we will include some new validation work that aims to better leverage the temporal information in the GLODAP dataset. In particular, this will focus on the largest spatial scales for which repeat hydrography is available in the North Atlantic.

B6 Reviewer 1 raises interesting questions about the role of preformed alkalinity in interannual variability. This manuscript certainly highlights that a detailed investigation of alkalinity variability in the North Atlantic is warranted. The inclusions of a closed budget analysis will shed some light on this point, but a thorough study is beyond the scope of this work and is left to future research. To address this point, however, more interpretation of our results will be included. We agree with the reviewer that biological variability is unlikely to dominate on these timescales (since soft tissue carbon variability is small). Concentration/dilution effects are likely to be important or leading order (this is why, for example, it is possible to estimate much of the interannual DIC variability using salinity as a predictor of preformed alkalinity). The effects of transport across the lateral boundary will be addressed with the inclusion of the closed budget analysis. However, other effects are also important. In this study, interannual variability in preformed alkalinity depends on 1) surface alkalinity (concentration-dilution effects due to the hydrological cycle primarily) and 2) the location and intensity of water subduction. It is not trivial to distinguish these processes, and would make a good focus for subsequent work.

---

## Author Comment (AC2) · 3 May 2019

Response to Reviewer 2

We would like to thank the reviewer for their detailed and insightful evaluation of our manuscript. Clearly the reviewer took time and care to critically engage with the work, as this is reflected in the constructive nature of the comments. We agree that the suggestions regarding some reorganisation of the manuscript will benefit the work.

Major comments: 1. We agree that Section 4 requires some restructuring, including removing the repetitive elements of Section 4.4, and Figure 7. It may be the case that this reorganisation improves the structure of the section sufficiently, and that separating sections by timescale may not be necessary. We will try both and see which is

preferable. 1a: Plotting the components of DIC in Figure 3 or in separate Figures is a tempting way to spot similarities, and our previous attempts at doing so have created crowded or a multitude of figures, as the reviewer suspects. The aim of including such figures (and the spectra in Figure 2b) was to show a more intense collection of information that would then be distilled down using the subsequent statistical analysis. Preliminary experimentation with re-plotting these data shows that it should be possible to produce a plot with a separate subpanel for each component at each timescale to clearly convey this information. 1b: We agree that the text on page 11 should be revised to better specify how Figure 4 implies each of these hypotheses. In addition, these hypotheses will be addressed more thoroughly throughout the manuscript (as each piece of analyses confirms or refutes them), and a summary of the hypotheses found to be accepted will be included at the end. This comment was common to both reviewers and we agree the text must be updated to address this. As for reproducing Figure 4 for additional components, we will experiment with this and see what balance can be struck between conveying useful versus excessive additional information. 1c: We agree that this section should be largely removed and the relevant parts folded into the rest of the manuscript.

2 We agree that this work should be reorganised within the manuscript following the reviewer's suggestion. The incorporation of the closed-budget analysis will address this. In particular, the correlation analysis would indeed benefit from being added into Section 5.

Minor comments: All the minor points made by Reviewer 2 are good ones, and we agree with the proposed suggestions.